# A spatial threshold for astrocyte calcium surge

**Justin Lines[1]\*, Andres Baraibar[1], Carmen Nanclares[1], Eduardo D Martin[2], Juan Aguilar[3], Paulo Kofuji[1], Marta Navarrete[2†], Alfonso Araque[1†]**

[1]Department of Neuroscience, University of Minnesota, Minneapolis, United States; [2]Instituto Cajal, CSIC, Madrid, Spain; [3]Experimental Neurophysiology Group, Hospital Nacional de Parapléjicos SESCAM, Toledo, Spain

## eLife Assessment

This study presents **valuable** findings that add to our understanding of cortical astrocytes, which respond to synaptic activity with calcium release in subcellular domains that can proceed to larger calcium waves. The proposed concept of a spatial "threshold" is based on **solid** evidence from in vivo and ex vivo imaging data and the use of mutant mice. Details of the specific threshold must be taken with caution and are necessarily incomplete, but may be supported by additional experiments with higher resolution in space and time in the future.

**\*For correspondence:**
justin.lines@mssm.edu

†These authors contributed equally to this work

**Competing interest:** The authors declare that no competing interests exist.

**Abstract** Astrocytes are active cells involved in brain function through the bidirectional communication with neurons, in which astrocyte calcium plays a crucial role. Synaptically evoked calcium increases can be localized to independent subcellular domains or expand to the entire cell, i.e., calcium surge. Because a single astrocyte may contact ~100,000 synapses, the control of the intracellular calcium signal propagation may have relevant consequences on brain function. Yet, the properties governing the spatial dynamics of astrocyte calcium remains poorly defined. Imaging subcellular responses of cortical astrocytes to sensory stimulation in mice, we show that sensory-evoked astrocyte calcium responses originated and remained localized in domains of the astrocytic arborization, but eventually propagated to the entire cell if a spatial threshold of >23% of the arborization being activated was surpassed. Using *Itpr2⁻/⁻* mice, we found that type-2 IP$_3$ receptors were necessary for the generation of astrocyte calcium surge. We finally show using in situ electrophysiological recordings that the spatial threshold of the astrocyte calcium signal consequently determined the gliotransmitter release. Present results reveal a fundamental property of astrocyte physiology, i.e., a spatial threshold for astrocyte calcium propagation, which depends on astrocyte intrinsic properties and governs astrocyte integration of local synaptic activity and subsequent neuromodulation.

## Introduction

Accumulating evidence indicates that astrocytes play active roles in synaptic function and neural information processing by exchanging signals with neurons. They respond to synaptic activity with intracellular calcium elevations, which stimulate the release of gliotransmitters that regulate neuronal and synaptic function (*Araque et al., 2014*; *Perea et al., 2009*). Hence, the astrocyte calcium signal is a crucial signaling event in the bidirectional communication between neurons and astrocytes. Moreover, astrocyte calcium manipulations have been shown to regulate neuronal network function (*Ahmadpour et al., 2024*; *Lee et al., 2014*; *Lines et al., 2020*; *Mederos et al., 2021*; *Miguel-Quesada et al., 2023*; *Perea et al., 2016*; *Poskanzer and Yuste, 2016*) and animal behavior (*Adamsky et al., 2018*; *Corkrum et al., 2020*; *Kofuji and Araque, 2021*; *Martin-Fernandez et al., 2017*; *Monai et al., 2016*;

*Nagai et al., 2021*; *Oliveira et al., 2015*), and disruption of astrocyte calcium has been proposed to contribute to brain diseases (*Delekate et al., 2014*; *Jiang et al., 2016*; *Kuchibhotla et al., 2009*; *Lines et al., 2022*; *Nanclares et al., 2023*; *Tian et al., 2005*; *Yu et al., 2018*).

Astrocyte calcium variations represent a complex signal that exists over a wide range of spatial and temporal scales (*Ahrens et al., 2024*; *Bazargani and Attwell, 2016*; *Rusakov, 2015*; *Semyanov et al., 2020*; *Shigetomi et al., 2016*; *Volterra et al., 2014*). Spatially, the astrocyte calcium signal may occur at discrete subcellular regions, termed domains, in the astrocyte arborization, or may encompass large portions of the cell or even the entire astrocyte (*Bazargani and Attwell, 2016*; *Rusakov, 2015*; *Semyanov et al., 2020*; *Shigetomi et al., 2016*; *Volterra et al., 2014*). Subcellular astrocyte calcium events have been recently proposed to be involved in the representation of spatiotemporal maps (*Curreli et al., 2022*; *Doron et al., 2022*; *Serra et al., 2022*) and to contribute to long-term information storage (*Curreli et al., 2022*; *Doron et al., 2022*; *Georgiou et al., 2022*; *Vignoli et al., 2021*). Calcium activity in astrocytes is believed to originate in domains within astrocytic processes that contact and bidirectly communicate with nearby synapses termed microdomains (*Arizono et al., 2020*; *Bindocci et al., 2017*; *Di Castro et al., 2011*; *Grosche et al., 1999*; *Khakh and Sofroniew, 2015*; *Panatier et al., 2011*). While calcium transients in discrete domains have been found to be independent events within astrocyte arborizations (*Chen et al., 2020*; *Stobart et al., 2018*; *Ung et al., 2021*), they can also occur in concert (*Agarwal et al., 2017*; *Georgiou et al., 2022*; *Otsu et al., 2015*; *Shigetomi et al., 2013*) and eventually expand to a larger cellular region, including the astrocytic soma, a phenomenon termed astrocyte calcium surge (*Hirase et al., 2004*).

Moreover, the regulation of the amplitude and spatial extension of the astrocyte calcium signal by the coincident activity of different synaptic inputs has been proposed to endow astrocytes with integrative properties for synaptic information processing (*Durkee and Araque, 2019a*; *Fedotova et al., 2023*; *Lines, 2025*; *Perea and Araque, 2005*; *Rupprecht et al., 2024*). Because a single astrocyte may contact ~100,000 synapses (*Bushong et al., 2002*), the integrative properties of a single astrocyte and the control of the intracellular calcium signal propagation may have relevant consequences by regulating the spatial range of astrocyte influence on synaptic terminals (*Fellin et al., 2004*; *Gordleeva et al., 2019*; *Perea and Araque, 2005*). While there have been recent works describing molecular underpinnings of microdomain calcium transients (*Diaz et al., 2019*; *Ma and Freeman, 2020*; *Montagna et al., 2019*), the underlying processes governing the connection between the two subcellular activity states – independent or concerted events – and the spatial extension of the intracellular calcium signal remain unknown.

To address these issues, we have monitored sensory-evoked astrocyte calcium activity in the mouse primary somatosensory cortex in vivo, combining astrocyte structural imaging data and subcellular imaging analysis. Here, we use an unbiased and semi-automatic algorithm to perform high-throughput analysis across a large number of astrocytes (~1000) to discover a subcellular property. We have found that astrocyte calcium responses originate in the surrounding arborizations and propagate to the soma if over 23% of the surrounding arborization is activated. If the astrocyte calcium spatial threshold is overcome, this spurs a surge of calcium into the surrounding arborization. Using transgenic *Itpr2*[-/-] mice, we found that the activation of type-2 IP$_3$ receptors is necessary for the generation of astrocyte calcium surge. Patch-clamp recordings of neurons near activated astrocytes showed an increase in slow inward currents (SICs), detailing an output of astrocyte calcium surge. Using a combination of structural and functional two-photon imaging of astrocyte activity, we define a fundamental property of astrocyte calcium physiology, i.e., a spatial threshold for astrocyte calcium propagation.

## Results

### Imaging astrocyte structure and function simultaneously in vivo

We simultaneously monitored calcium activity in identified sulforhodamine 101 (SR101)-labeled astrocytes in the primary somatosensory cortex using two-photon microscopy in vivo. We used transgenic mice expressing GCaMP6f in astrocytes, generated as described in the Methods (*Bindocci et al., 2017*; *Lines et al., 2020*), to monitor sensory-evoked intracellular astrocyte calcium dynamics in combination with SR101 labeling to monitor astrocyte morphology (*Nimmerjahn et al., 2004*; *Figure 1A and B*). Regions of interest (ROIs) were computationally determined from SR101-positive structural imaging (*Bindocci et al., 2017*) by outlining individual astrocytes and performing semi-automatic segmentation

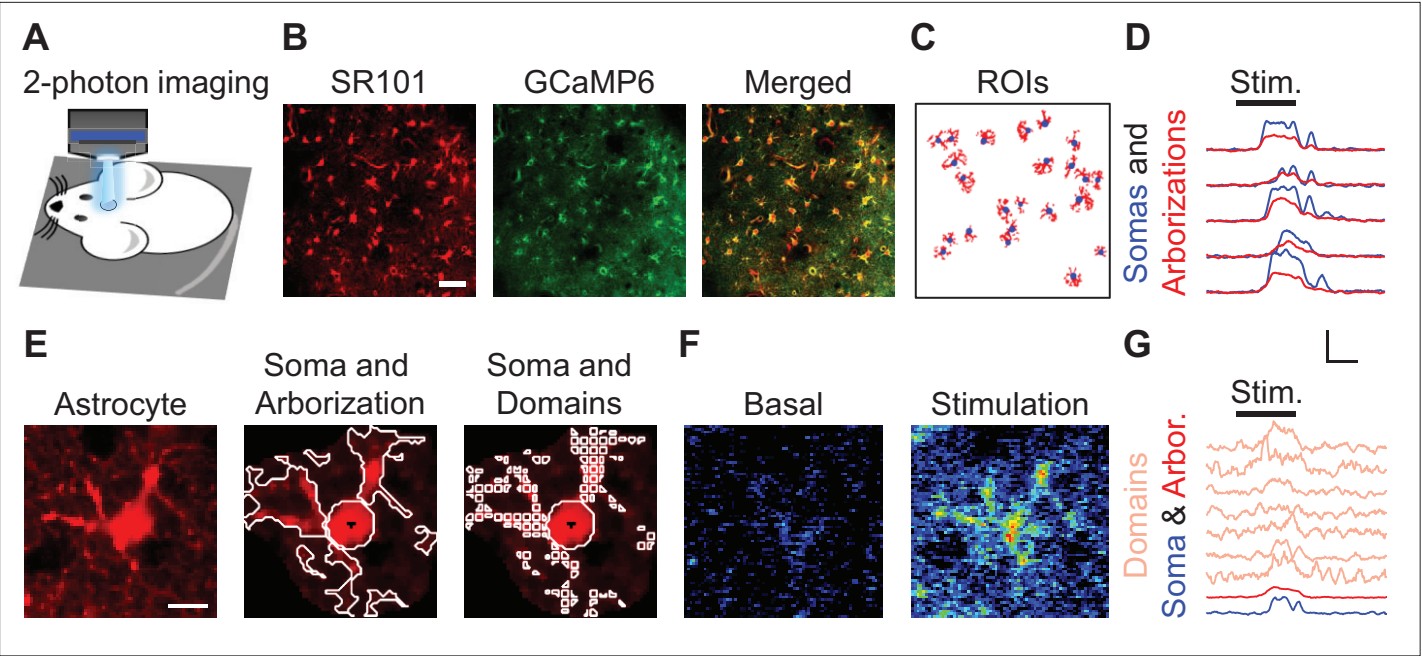

**Figure 1.** Imaging astrocyte structure and function simultaneously in vivo. (**A**) Scheme of in vivo preparation to image astrocyte Ca²⁺ and structure. (**B**) Sulforhodamine 101 (SR101)-stained astrocyte structure, GCaMP6 to monitor astrocyte Ca²⁺ signal, and merge. Scale bar = 50 μm. (**C**) Regions of interest (ROIs) from SR101-stained structure of somas (blue) and arborizations (red). (**D**) Ca²⁺ traces from B from somas (blue) and arborizations (red). Scale = $F/F_o$, 10 s. (**E**) SR101-stained astrocyte (left), ROIs outlining soma and arborization (center) and ROIs defining the soma and domains (right). Scale bar = 10 μm. (**F**) Pseudocolor Ca²⁺ image during basal (left) and hindpaw electrical stimulation (right). (**G**) Ca²⁺ traces from F from domains (salmon), arborization (red), and soma (blue). Scale = $F/F_o$, 10 s.

The online version of this article includes the following figure supplement(s) for figure 1:

**Figure supplement 1.** Semi-automatic method for the segmentation of astrocyte morphology.

into somas and arborizations (*Figure 1C*; see *Figure 1—figure supplement 1* for an in-depth description of segmentation). Next, subcellular quantification of soma and arborization calcium signals from individual astrocytes was evaluated in response to peripheral electrical stimulation of the hindpaw (2 mA at 2 Hz for 20 s; *Figure 1D*). Following the segmentation of an individual astrocyte into soma and arborization, the astrocyte arborization was further discretized into a grid of maximally sized 4.3 × 4.3 μm² square regions of interest, which we define as astrocyte domains (*Figure 1E and F*; *Agarwal et al., 2017*; *Di Castro et al., 2011*; *Grosche et al., 1999*; *Shigetomi et al., 2013*). The concept of domain to define all subcellular domains in the astrocyte arborization should not be confused with the concept of microdomain, that usually refers to the distal subcellular domains in contact with synapses. Thus, we were able to quantify the sensory-evoked calcium responses in individual domains, as well as in the arborization and soma (*Figure 1G*).

The analysis of the sensory-evoked calcium activity from astrocyte arborization and soma uncovered that (1) the majority of responses occurred in both the soma and arborization (57.7 ± 4.5%; n=30 populations, 3 animals); (2) some responses occurred only in the arborization (15.1 ± 1.6%; n=30 populations, 3 animals; *Figure 2A–D*); (3) a small minority of responses included activity in the soma but not the arborization (3 ± 0.5%; n=30 populations, 3 animals; *Figure 2D*); and (4) some astrocytes did not respond (24.1 ± 3.6%; n=30 populations, 3 animals). Since the majority of cells showed responses in both the soma and arborization, we hypothesized that the proportion of domain activity within the arborization and the somatic calcium activity were correlated across a population of astrocytes. To test this hypothesis, we first examined the average percentage of responding arborizations vs the percentage of soma activation within a population and found a significant linear correlation between these subcellular measures of activity (linear correlation: p<0.001, R²=0.90; n=30 populations, 3 animals; *Figure 2E*). Additionally, averaging the percentage of active domains per cell over a population vs the percentage of somas active showed a significant linear correlation (linear

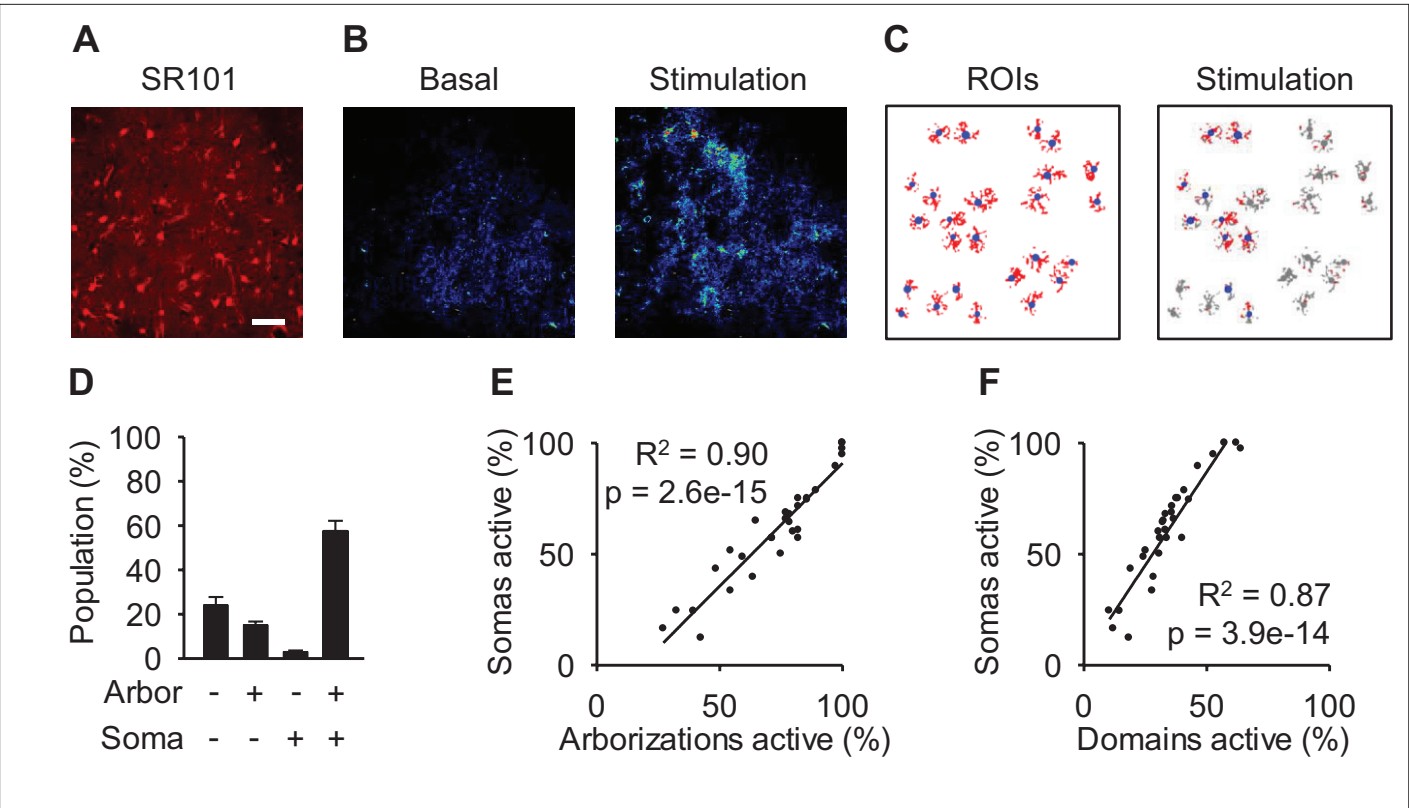

**Figure 2.** Population arborization calcium is correlated to population soma activity. (**A**) Sulforhodamine 101 (SR101) staining. Scale bar = 50 µm. (**B**) Pseudocolor Ca²⁺ images at basal and stimulation. (**C**) Regions of interest (ROIs) of soma and arborizations/domains along with activity during stimulation. (**D**) Proportion of subcellular responses to stimulation. (**E**) Percentage of active arborizations vs percent of somas active. (**F**) Percentage of domains active vs percent of somas active. Mean ± SEM. Pearson correlation.

correlation: p<0.001, $R^2$=0.87; *n*=30 populations, 3 animals; *Figure 2F*). These results indicate that, on average, subcellular calcium events located in astrocyte arborizations are related to soma activation.

### Subcellular astrocyte calcium originates in the arborization

We then analyzed the spatial and temporal properties of the intracellular calcium dynamics in astrocytic somas and arborizations (*Figure 3A–D*). First, we determined the kinetics of the sensory-evoked astrocyte calcium signal. Sensory-evoked calcium rises in arborizations occurred with a delay of 11.1±0.3 s from the onset of the peripheral stimulation and significantly preceded those occurring in the soma with a 13.2±0.2 s delay from stimulus onset (p<0.001; *n*=30 populations, 3 animals; *Figure 3D–F*). Moreover, rise time to peak and decay time back to baseline of the calcium traces were faster in somas than arborizations (10–90% rise time: 5.7±0.2 s in arborizations vs 3.5±0.2 s in somas; p<0.001; 90–10% decay time: 4.8±0.2 s in arborizations vs 4.3±0.2 s in somas; p<0.01; *n*=30 populations, 3 animals; *Figure 3E and F*). These results indicate that astrocyte responses occurred initially in the arborizations, which is consistent with the idea that synapses are likely to be accessed at the astrocyte arborization (*Arizono et al., 2020*; *Papouin et al., 2017*).

### A spatial threshold to activate the soma and calcium surge

Next, we determined the relative spatial relationship of calcium activity of domains within the arborizations and somas of individual astrocytes (*Figure 4A*). We quantified the proportion of subcellular domains in individual astrocytic arborizations that responded to electrical stimuli with varied parameters and assessed whether the corresponding soma responded (*Figure 4A–C*). When changing the stimulus parameters (duration, frequency, and intensity), the number of responding domains increased as the stimulus duration, frequency, and intensity increased (ANOVA: duration: p<0.001, frequency: p<0.001, intensity: p<0.001; *n*=11 populations, 4 animals; *Figure 4D*). As described

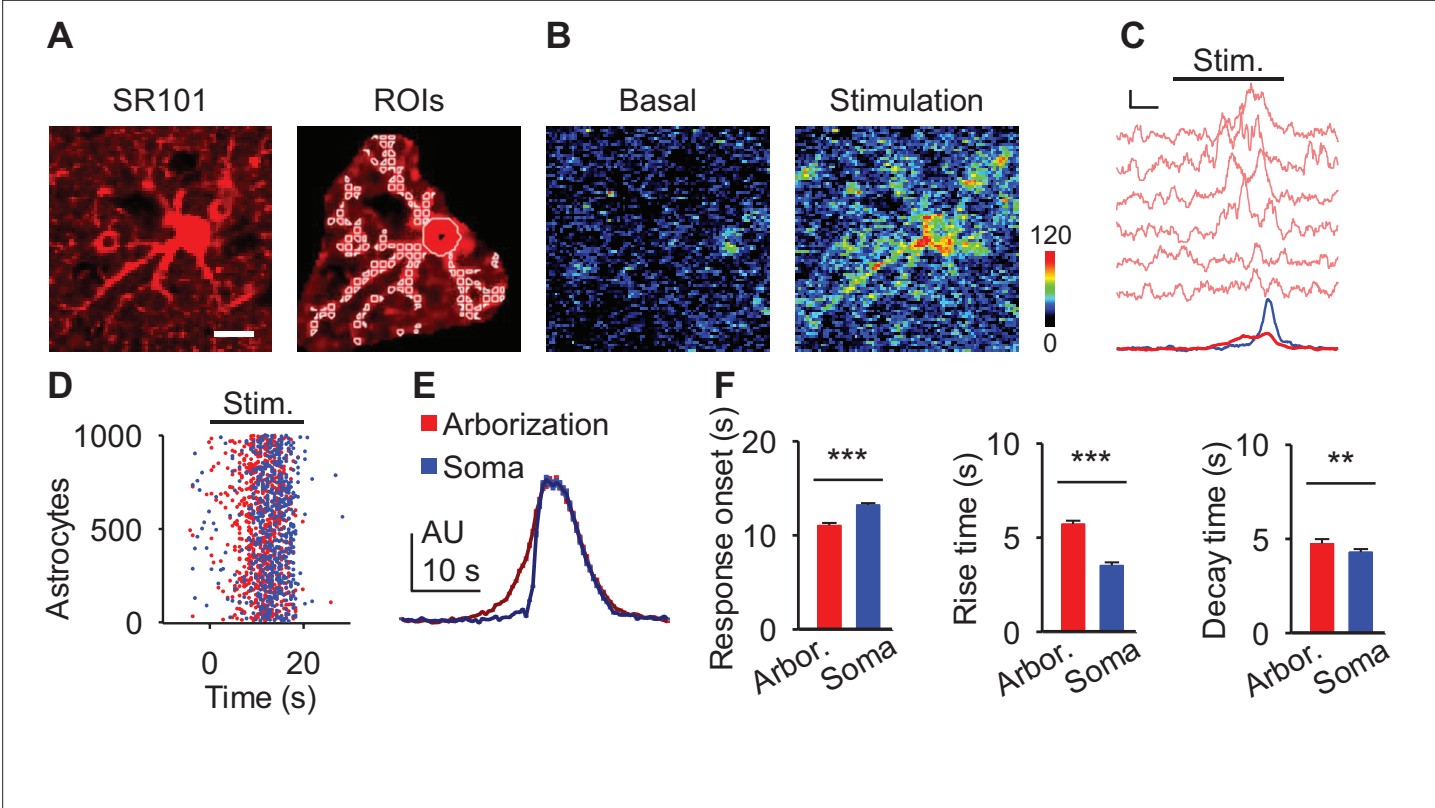

**Figure 3.** Astrocyte calcium responses originate in the arborization before the soma. (**A**) Astrocyte with regions of interest (ROIs). Scale bar = 10 μm. (**B**) Pseudocolor Ca²⁺ image. (**C**) Ca²⁺ traces in B from domains (pink), arborization (red), and the soma (blue). Scale = $F/F_o$, 5 s. (**D**) Raster plot of astrocyte somas (blue) and arbors (red) in response to stimulation (gray). (**E**) Average calcium traces from somas (blue) and arborizations (red) aligned to their respective soma onset. (**F**) Soma and arbor latency to response (left), event rise time (center), and event decay time (right). n = 995 astrocyte reponses. Mean ± SEM. '**' ≡ p<0.01 and '***' ≡ p<0.001 using paired Student's t-test.

The online version of this article includes the following figure supplement(s) for figure 3:

**Figure supplement 1.** Distinct dynamics of domains before and after soma onset.

above, the probability of soma activation vs the percentage of active domains could be accurately fit to a linear regression (see *Figure 2F*) indicating a correlation between these variables. To further characterize this relationship, we plotted paired values of on/off active soma (i.e. activated or not) vs the proportion of active domains from individual astrocytes. We found that the activation of a relatively low proportion of domains occurred without activation of the soma (*Figure 4E*). Conversely, large proportions of activated domains were accompanied by a calcium elevation in the soma (*Figure 4E*). This relationship suggested the existence of a threshold. Fitting these values to the Heaviside step function (in Methods, *Equation 4*; *Davies, 2002*) indicated that somas were active when at least 22.6% of their respective domains were active ($R^2$=0.42; n=995 astrocytes from 30 populations and 3 animals; *Figure 4E*). This spatial threshold value was independent of the sensory input because similar values were found across various stimulus parameters (one-way ANOVA: duration: p=0.50, frequency: p=0.29, intensity: p=0.38; n=11 populations, 4 animals; *Figure 4F*), suggesting that it is determined by intrinsic astrocyte properties. Consolidating spatial threshold measurements from various stimulation parameters, we quantified the spatial threshold to be within 95% confidence intervals of [21.2%, 24.0%]. Moreover, plotting the percent of active domains for an individual astrocyte vs the amplitude of the somatic calcium response was fit to a sigmoid curve ($R^2$=0.58; n=995 astrocytes from 30 populations and 3 animals; *Figure 4G*). These fits to cellular data as well as the large cluster of unchanged somatic amplitude with subthreshold domain activity further confirms that nonresponsive somas were not just below event detection, but indeed the soma does not become active. Taken together, these results indicate the existence of a spatial threshold for soma activation determined by astrocyte intrinsic properties and the proportion of active domains.

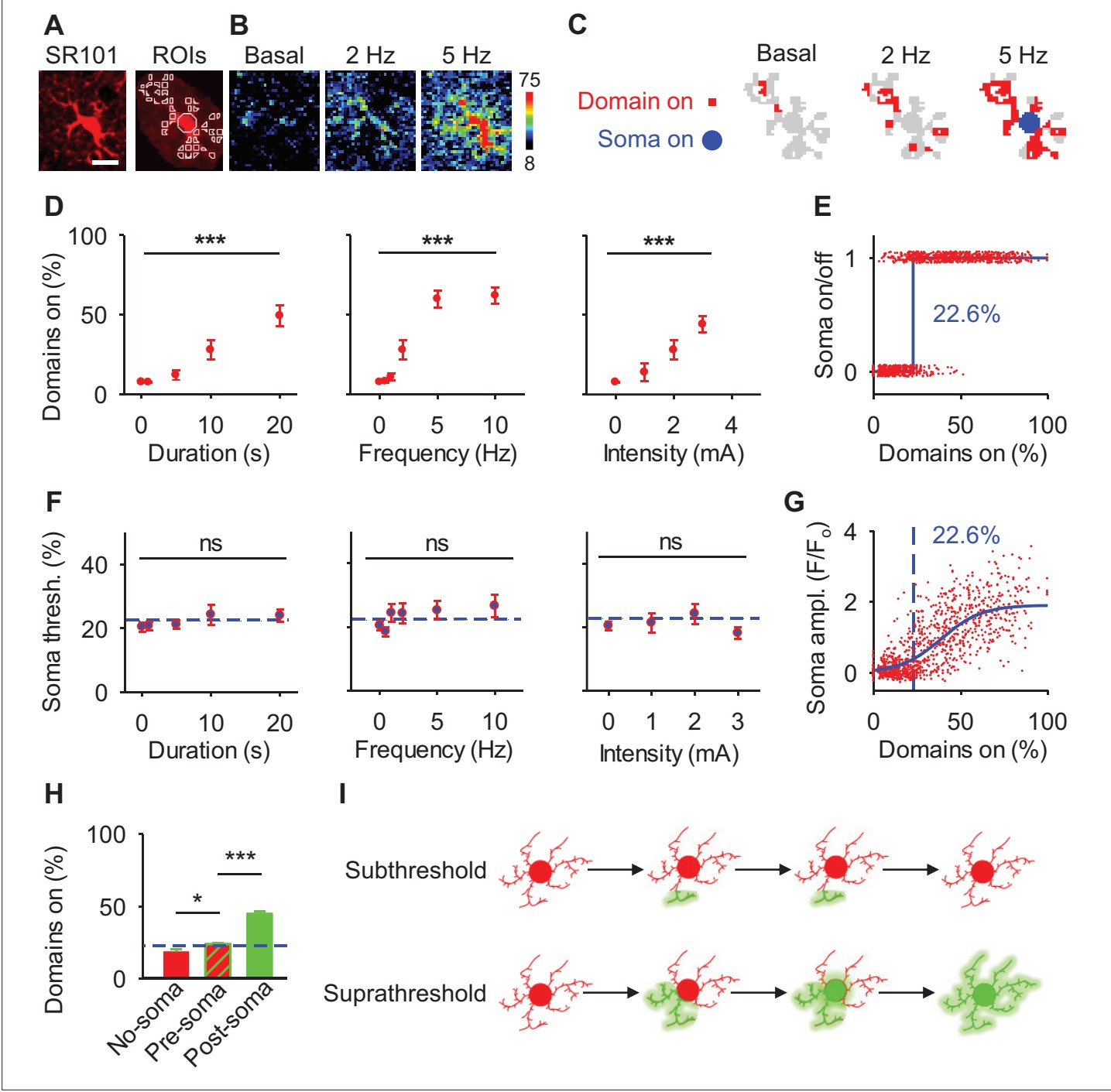

**Figure 4.** A spatial threshold for astrocyte calcium in the soma to reach astrocyte calcium surge. (**A**) Astrocyte and regions of interest (ROIs). Scale bar = 10 μm. (**B**) Pseudocolor $Ca^{2+}$ images during basal and different frequency of stimulations. (**C**) Scheme of domains (red) and soma (blue) $Ca^{2+}$ activity from B. (**D**) Percentage of active domains vs stimulus duration, intensity, and frequency. n = 11 populations. (**E**) Active state of soma for individual astrocytes vs percentage of active domains (red). Data were fit to a Heaviside step function (blue dotted line). (**F**) Percentage of active domains necessary to elicit soma activation vs stimulus duration, intensity, and frequency. Blue dotted lines denote 22.6% spatial threshold. n = 11 populations. (**G**) Soma fluorescence vs percentage of active domains (red). Data were fit to a sigmoidal function (blue) and a blue dotted line denotes 22.6% spatial threshold. (**H**) Percentage of active domains in the absence of soma activation vs active domains before and after soma activation. Blue line denotes 22.6% spatial threshold. (**I**) Schematic showing subthreshold and suprathreshold astrocyte calcium activity. Mean ± SEM. '***' ≡ p<0.001 and 'ns' ≡ p>0.05 using one-way ANOVA or Student's t-test.

The presence of a spatial threshold for somatic responses suggests that cells that respond with soma activity would have a higher response of domains prior to the soma response (pre-soma) when compared to astrocytes without a soma response (no-soma). Indeed, when comparing these populations we found that astrocytes with responding somas had a significantly larger proportion of surrounding domains active prior to soma activation (pre-soma) when compared to cells without a somatic response (no-soma), confirming our hypothesis of a spatial cellular threshold ($18.5 \pm 1.7\%$ of active domains in no-soma vs $23.9 \pm 0.8\%$ of domains in pre-soma cells, $p<0.05$; $n=607$ active vs $n=388$ not active, 30 populations and 3 animals; *Figure 4H*). Further, we found that astrocytes with an excess of 22.6% of domain activity, that induced a somatic response, led to increased domain activation throughout the remaining arborization (post-soma), i.e., domain activity before soma activation (pre-soma) vs domain activity after soma activation (post-soma) ($23.9 \pm 0.8\%$ in pre-soma firing compared to $45.0 \pm 1.8\%$ in post-soma activation, $p<0.001$; $n=607$ astrocytes, 30 populations in 3 animals; *Figure 4H*, *Figure 3—figure supplement 1*). Together, these results confirm that astrocytes that respond with domain activity in excess of the spatial threshold precipitates somatic activity and a calcium surge of expanded responses throughout the astrocytic arborization (*Figure 4I*).

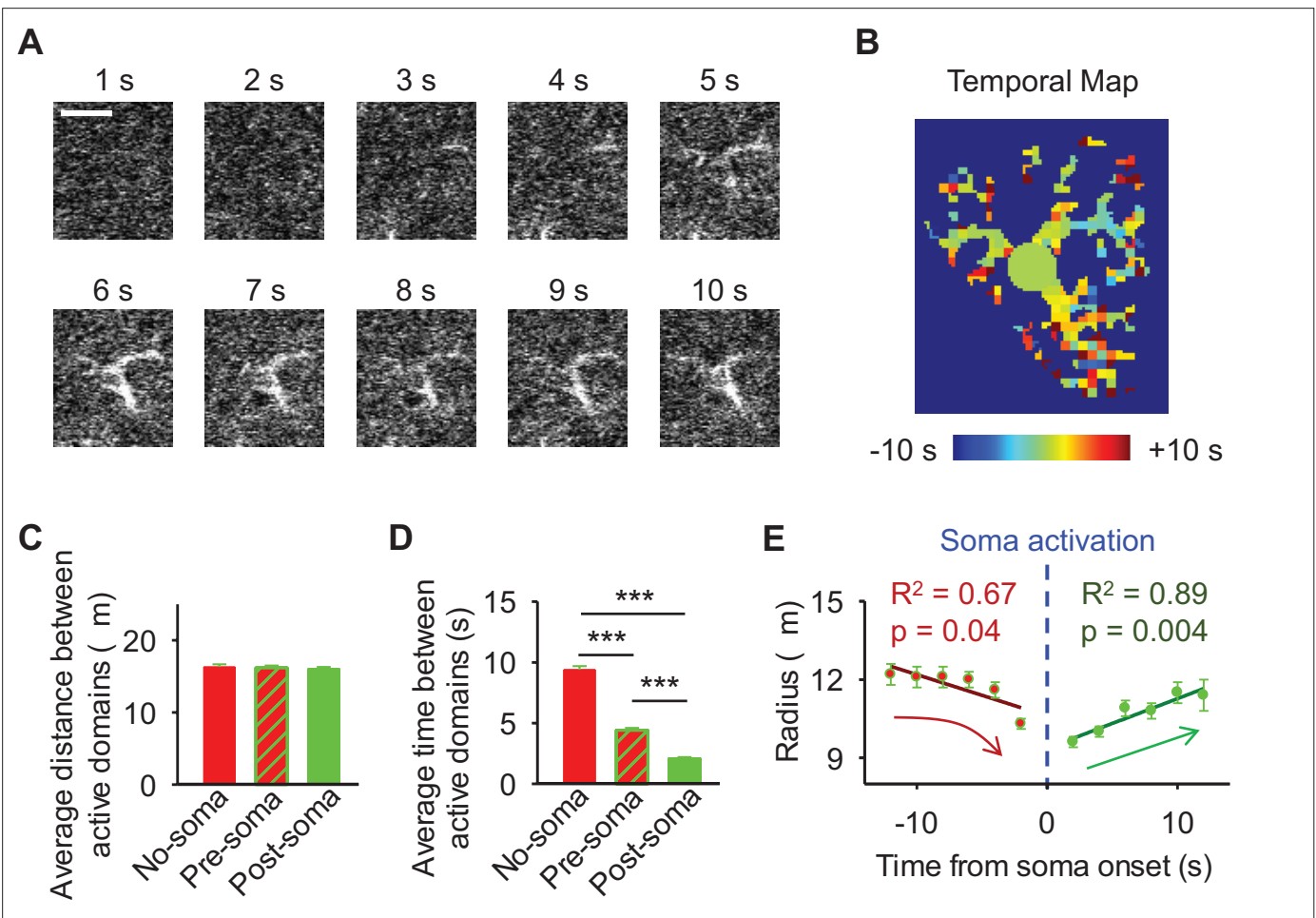

**Figure 5.** The spatiotemporal properties of astrocyte calcium surge. (**A**) The time course of calcium surge following sensory stimulation. The average of 10 sequential frames (2 s) ending on the time from stimulation onset in seconds. Scale bar = 10 μm. (**B**) Temporal map of domain temporal order relative to soma activation from A. (**C**) Average distance between pairs of active domains in the absence of soma activation vs before and after soma activation. n = 286 cells without soma activation, and 326 cells with soma activation. (**D**) Average time between pairs of active domains in the absence of soma activation vs before and after soma activation. Mean ± SEM. (**E**) Comparison between activated domain onset relative to soma activation vs radius from the center of the soma. '***' ≡ p<0.001 using paired and unpaired Student's t-test. Pearson correlation.

## The spatiotemporal characteristics of active domains in astrocyte calcium responses

Our results demonstrate the relationship between the percentage of active domains and soma activation and subsequent calcium surge. Next, we were interested in the spatiotemporal properties of domain activity leading up to and during calcium surge. Because we imaged groups of astrocytes, we were able to constrain our analyses to early responders (onset<median population onset) in order to evaluate astrocytes that were more likely to respond to neuronal-evoked sensory stimulation and not nearby astrocyte activation (*Figure 5A*). In this population the spatial threshold was 23.8% within the 95% confidence intervals of [21.2%, 24.0%]. First, we created temporal maps, where each domain is labeled as its onset relative to soma activation, of individual astrocyte calcium responses to study the spatiotemporal profile of astrocyte calcium surge (*Bindocci et al., 2017*; *Figure 5B*). Using temporal maps, we quantified the spatial clustering of responding domains by measuring the average distance between active domains. We found that the average distance between active domains in subthreshold astrocyte responses was not significantly different from pre-soma suprathreshold activity (16.3±0.4 µm in no-soma cells vs 16.2±0.3 µm in pre-soma cells, p=0.75; n=286 no-soma vs n=326 pre-soma, 30 populations and 3 animals; *Figure 5C*). Following soma activation, astrocyte calcium surge was marked with no significant change in the average distance between active domains (16.0±0.3 µm in post-soma cells vs 16.3±0.4 µm in no-soma cells, p=0.57 and 16.2±0.3 µm in pre-soma cells, p=0.31; n=326 soma active and n=286 no soma active, 30 populations and 3 animals; *Figure 5C*). Taken together this suggests that on average domain activation may happen in a nonlocal fashion, possibly illustrating the underlying nonlocal activation of nearby synaptic activity. Next, we interrogated the temporal patterning of domain activation by quantifying the average time between domain responses, and found that the average time between domain responses was significantly decreased in pre-soma suprathreshold activity compared to subthreshold activities without subsequent soma activation (9.4±0.3 s in no-soma cells vs 4.4±0.2 s in pre-soma cells, p<0.001; n=326 soma active vs n=286 not soma active, 30 populations and 3 animals; *Figure 5D*). The average time between domain activation was even less after the soma became active during calcium surge (2.1±0.1 s in post-soma vs 9.4±0.3 s in no-soma cells, p<0.001 and 4.4±0.1 s in pre-soma cells, p<0.001; n=326 soma active and n=286 not soma active, 30 populations and 3 animals; *Figure 5D*). This corroborates our findings in *Figure 3—figure supplement 1* and highlights the difference in temporal profiles between subthreshold activity and astrocyte calcium surge.

We then tested the contribution of each of our three variables describing domain activation (percent area, average distance, and time) to elicit soma activation by creating a general linear model. We found that overall, there was a significant relationship between these variables and the soma response (p=5.5e-114), with the percent area having the largest effect (p=3.5e-70) followed by the average time (p=3.6e-7), and average distance having no significant effect (p=0.12). Taken together this suggests that the overall driver of soma activation appears to be driven by the percent area of active domains within a constrained time window.

Recent work studying astrocyte integration has suggested a centripetal model of astrocyte calcium, where more distal regions of the astrocyte arborization become active initially and activation flows toward the soma (*Fedotova et al., 2023*; *Rupprecht et al., 2024*). Here, we confirm this finding, where activated domains located distal from the soma respond sooner than domains more proximal to the soma (linear correlation: p<0.05, $R^2$=0.67; n=30 populations, 3 animals; *Figure 5E*). Next, we build upon this result to also demonstrate that following soma activation, astrocyte calcium surge propagates outward in a centrifugal pattern, where domains proximal to the soma become active prior to distal domains (linear correlation: p<0.01, $R^2$=0.89; n=30 populations, 3 animals; *Figure 5E*). Together these results detail that intracellular astrocyte calcium follows a centripetal model until the soma is activated leading to a calcium surge that flows centrifugally. This suggests that astrocytes have the capabilities to integrate the nearby local synaptic population, and if this activity exceeds the spatial threshold then it leads to a whole-cell response that spreads outward.

## The type-2 IP$_3$ receptor is necessary for astrocyte calcium surge

Previous reports have demonstrated in transgenic mice with type-2 IP$_3$ receptors knocked out (*Itpr2$^{-/-}$* mice) have ablated somatic calcium activity, but preserved certain domain calcium events (*Agarwal et al., 2017*; *Lines et al., 2020*; *Schmidt and Oheim, 2020*; *Srinivasan et al., 2015*). Since

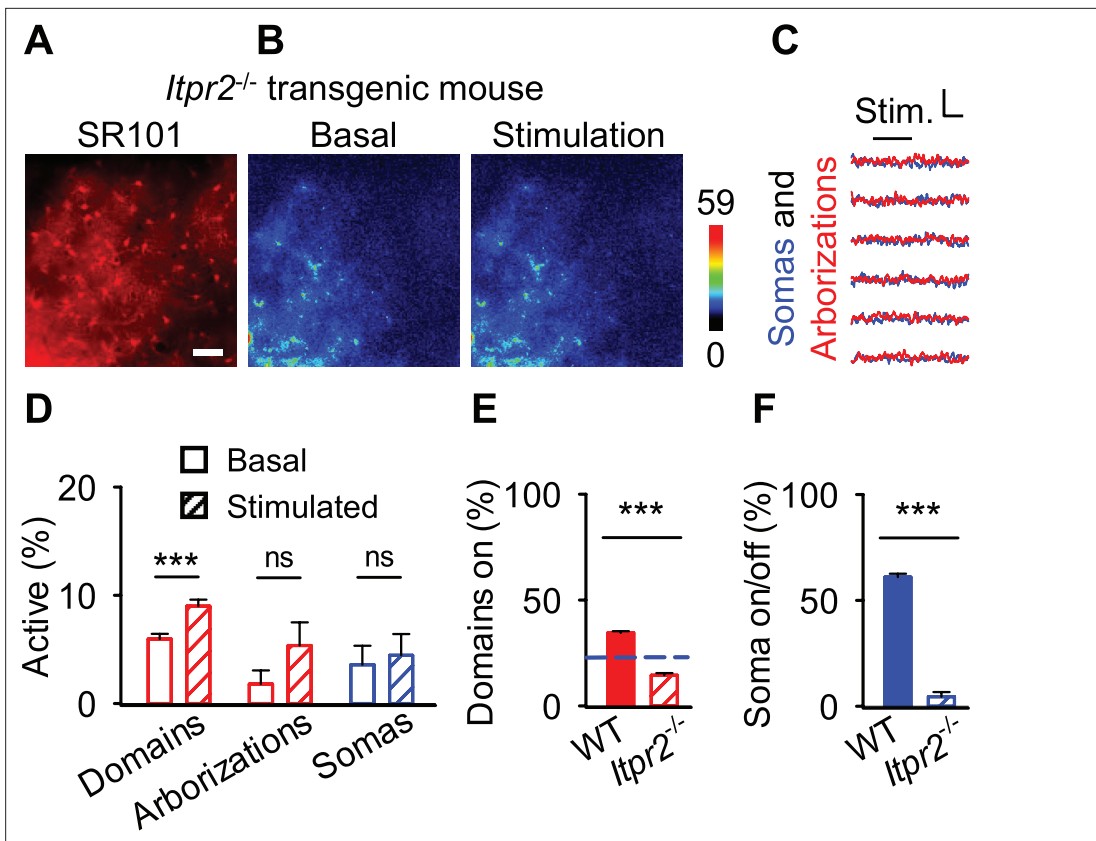

**Figure 6.** The spatial activation of domain $Ca^{2+}$ remains below the spatial threshold in mice lacking the $IP_3$ receptor type-2. (**A**) Sulforhodamine 101 (SR101) staining. Scale bar = 50 µm. (**B**) Pseudocolor $Ca^{2+}$ images at basal and stimulation. (**C**) Traces from astrocytes in B. Scale = $F/F_o$, 10 s. (**D**) Percentage of domains (left), arborizations (center), and somas (left) active at basal (open) and stimulation (hashed) in $Itpr2^{-/-}$ mice. (**E**) Percentage of domains active in wildtype (filled) and $Itpr2^{-/-}$ mice (hashed). Blue line denotes 22.6% spatial threshold. (**F**) Probability of soma activation in wildtype (filled) and $Itpr2^{-/-}$ mice (hashed). n = 30 populations in WT and 5 populations in knockout mice. Mean ± SEM. '***' ≡ p<0.001 using paired and unpaired Student's t-test.

$IP_3R2$-mediated calcium mobilization is an important signaling pathway in astrocyte calcium dynamics (*Guerra-Gomes et al., 2017*; *Lim et al., 2021*), we hypothesized that $IP_3R2$ activity was necessary for astrocyte calcium surge. To test this, we injected an adeno-associated virus to express GCaMP6f within astrocytes under the astroglial GfaABD1d promoter (AAV-GfaABC1d-GCaMP6f, please see Methods) into the primary somatosensory cortex of $Itpr2^{-/-}$ mice. We then quantified the calcium activity within GCaMP6f-expressing SR101-labeled cortical astrocytes before and after sensory stimulation (2 mA, 2 Hz for 20 s; *Figure 6A–C*). In agreement with previous results (*Agarwal et al., 2017*; *Lines et al., 2020*; *Srinivasan et al., 2015*; *Stobart et al., 2018*), astrocytes in $Itpr2^{-/-}$ mice responded to stimulation within the domains, but not the arborizations (i.e. average signal over the entire astrocyte arborization) or the somas (in domains: 6.0 ± 0.5% in basal vs 9.0 ± 0.6% in stimulation, p<0.001; n=2450 domains; in arborizations: 1.8 ± 1.3% in basal vs 5.4 ± 2.1% in stimulation, p=0.15; n=112 arborizations; in somas: 3.6 ± 1.8% in basal vs 4.5 ± 2.0% in stimulation, p=0.74; n=112 somas, 5 populations in 2 animals; *Figure 6D*). Moreover, within individual astrocytes, the percentage of activated domains in $Itpr2^{-/-}$ mice in response to stimulation was reduced compared to wildtype mice (34.5 ± 0.8% in wildtype mice vs 14.5 ± 1.0% in $Itpr2^{-/-}$ mice, p<0.001; n=995 astrocytes in 30 populations in 3 wildtype mice vs n=112 astrocytes in 5 populations in 2 $Itpr2^{-/-}$ mice; *Figure 6E*). Notably, while domain activity in $Itpr2^{-/-}$ mice increased upon stimulation, the level of activation remained below the defined spatial threshold of 22.6% (*Figure 6E*; dashed blue line), which astrocytes in $Itpr2^{-/-}$ mice were unable to overcome. Further confirming this, the probability of astrocyte somatic responses to stimulation was dramatically reduced in $Itpr2^{-/-}$ mice compared to wildtype mice (61.0 ± 1.6% in wildtype mice vs 4.5 ± 2.2% in $Itpr2^{-/-}$ mice, p<0.001; n=995 somas in 30 populations in 3 wildtype mice vs n=112 somas in 5 populations in 2 $Itpr2^{-/-}$ mice; *Figure 6F*). Taken together, these results indicate that

IP$_3$R2-mediated calcium internal release is necessary for astrocyte calcium surge and further support the idea of the spatial threshold for astrocyte calcium spread.

## Astrocyte calcium surge is associated with gliotransmission

We finally investigated whether the spatial threshold for astrocyte calcium was related to gliotransmission. We performed patch-clamp recordings of layer 2/3 cortical neurons in cortical brain slices to monitor the NMDAR-mediated SICs, a biological assay of glutamate gliotransmission (*Araque et al.,*

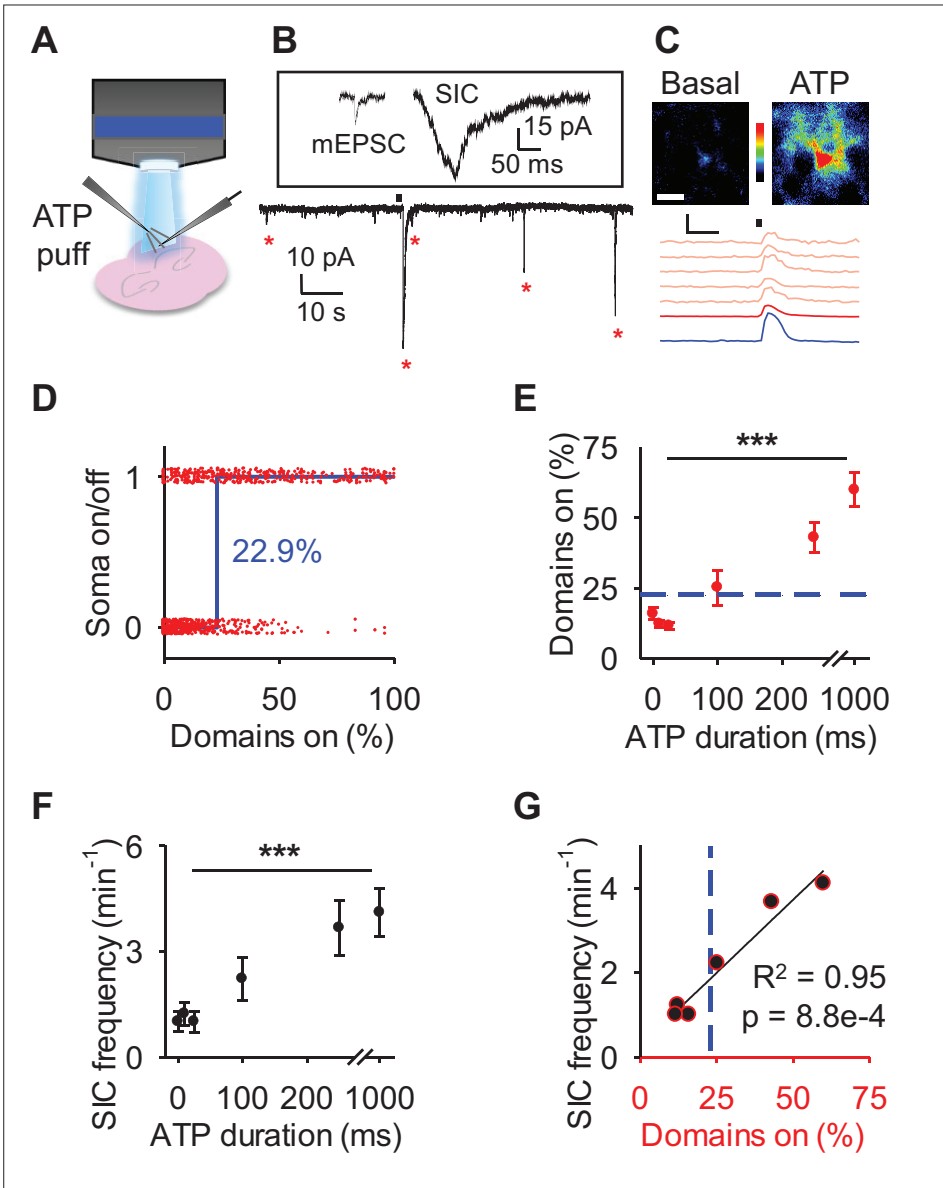

**Figure 7.** Increases in slow inward currents (SICs) occur with astrocyte calcium surge. (**A**) Scheme of cortical brain slice experiments to image astrocyte Ca$^{2+}$ and record SICs with adenosine triphosphate (ATP) application. (**B**) Example traces of a miniature excitatory post synaptic current (mEPCS) and an SIC (upper) and SICs following ATP puff (black bar) (lower). (**C**) Pseudocolor Ca$^{2+}$ images at basal and ATP with traces of responses to puff (black bar) in the soma (blue), arbor (red), and domains (salmon). Scale bar = 10 μm. Scale = $F/F_o$, 10 s. (**D**) Active state of soma for individual astrocytes vs percentage of active domains (red), with fit to a Heaviside step function (blue line). (**E**) Percentage of domains active in response to ATP puff. Blue dotted line denotes spatial threshold from D. n = 11 populations. (**F**) SIC frequency in response to ATP puff. n = 9 neurons. (**G**) Pearson correlation between percent active domains vs SIC frequency. Blue dotted line denotes spatial threshold from D. Mean ± SEM. '***' ≡ p<0.001 using one-way ANOVA and t-test of Pearson correlation.

*2000*; *Gómez-Gonzalo et al., 2018*) and applied different amounts of adenosine triphosphate (ATP) from a local micropipette with pressure pulses of different durations to gradually activate astrocytes (*Figure 7A–C*). Fluorescence imaging of astrocyte calcium in brain slices confirmed the existence of an astrocyte spatial threshold for calcium surge (22.9%; *Figure 7C and D*), that is within 95% confidence of our in vivo quantification [21.2%, 24.0%]. Beyond the threshold, increasing the duration of ATP puffs increased the proportion of activated astrocytic domains (one-way ANOVA: p<0.001, *n*=11 populations, 7 animals; *Figure 7E*; blue line indicates the threshold value obtained in *Figure 7D*). Likewise, similar to the domain activation, the SIC frequency increased as the duration of ATP puffs increased (one-way ANOVA: p<0.001, *n*=9 neurons, 9 animals; *Figure 7F*). Moreover, SIC frequency correlated with astrocyte domain activity (Pearson correlation: p<0.001, $R^2$=0.95; *Figure 7G*) but SIC frequency increased only beyond the spatial threshold of the astrocyte calcium signal (blue line in *Figure 7G*), indicating that the spatial threshold of the astrocyte calcium is correspondingly associated with gliotransmitter release. These results indicate that spatial threshold of the astrocyte calcium surge has a functional impact on gliotransmission, e.g., SICs have been found to be calcium-dependent related to the soma calcium and thus calcium surge (*Araque et al., 2000*), which have important consequences on the spatial extension of the astrocyte-neuron communication and synaptic regulation.

## Discussion

In the present study, we imaged calcium activity in identified SR101-labeled astrocytes of the primary somatosensory cortex in vivo, and developed and used an unbiased computational algorithm to integrate these data at different cellular levels, i.e., domains, arborizations, and somas. Here, we show that sensory-evoked astrocyte calcium responses originated in the arborization and were followed by delayed soma activation. A detailed examination of the domains within arborizations uncovered a correlation between domain activity and soma responses, and we were able to quantify a spatial threshold of activated domains necessary to produce soma activation (~23%). Domain activation was found to be stimulus dependent, however the spatial threshold for somatic response remained unchanged to various stimulus parameters, indicating that spatial threshold was determined by astrocytic intrinsic properties rather than synaptic inputs. We also found that soma responses preceded an increase in the spread of intracellular calcium activation across the arborization (i.e. calcium surge). Characterizing the spatiotemporal properties of astrocyte calcium in first responders, we detailed that temporal clustering was different across calcium surge, but not for spatial clustering. Further, we found centripetal calcium dynamics leading from the arborization to the soma to elicit calcium surge and centrifugal calcium spread. In *Itpr2*-/- mice, we found sensory-evoked calcium responses in astrocyte domains, albeit significantly reduced compared to wildtype mice and never reaching the defined spatial threshold to spur somatic activation. Finally, in cortical brain slices, we found that astrocyte calcium surge is related to nearby neuronal modulation as seen in the presence of SICs. Anesthesia has been shown to reduce astrocyte activity (*Thrane et al., 2012*), however we suppose that subcellular machinery is intact, and this is further supported in our slice experiments void of anesthesia. These results demonstrate that astrocytic responses to synaptic inputs were initiated in arborizations, extend intracellularly after reaching a spatial threshold of concomitant domain activation reliant on $IP_3R2$, and impact astrocyte to neuronal signaling.

Our results support the idea that neurotransmitters released at tripartite synapses act on microdomains at astrocytic arborizations (*Di Castro et al., 2011*; *Lia et al., 2021*; *Otsu et al., 2015*; *Panatier et al., 2011*) because they responded to sensory stimulation prior to astrocyte somas. Reports have found astrocyte domains can become active independently without recruiting neighboring arborizations or the soma, and domains also can become active en masse (*Agarwal et al., 2017*; *Shigetomi et al., 2013*). Our findings add to this, defining a spatial threshold of domains that needs to be reached in order to lead to soma activation and a calcium surge that propagates to the rest of the astrocyte arborization.

Several lines of evidence indicate that the spatial threshold does not result from increased stimulus parameters. First, the stimulus dependence of domain activity shows continuous activation with no threshold. Second, the spatial threshold identified using the Heaviside step function depends on domain activation and not stimulus parameters. Third, the spatial threshold is independent of the stimulus parameter, including no stimulation. Finally, using a different set of experiments in slice

recreated the spatial threshold. Overall, this evidence indicates the existence of a spatial threshold that is determined by intrinsic properties of the astrocyte.

Present results indicate that if the activation of a spatially localized astrocyte arborization by a localized set of synapses reaches the spatial threshold, the calcium signal is then globally expanded to modulate different synapses and neurons. The present demonstration of a spatial threshold to astrocyte calcium surge suggests that astrocytes spatially integrate information from multiple synaptic inputs. While previous reports have shown information integration of different neurotransmitters and synaptic inputs by astrocytes in situ (*Durkee et al., 2019b*; *Perea and Araque, 2005*), which may coordinate networks of neurons in silico (*Gordleeva et al., 2019*), our findings reveal novel integrative properties of spatial information by astrocytes in vivo.

Close examinations of the calcium surge uncovered distinct propagations whether before or after soma activation. First, our analysis found that temporal clustering changed before and after calcium surge, with both being above subthreshold activity, and that this characteristic was absent when assessing spatial clustering. When comparing the percent area, spatial and temporal clustering of active domains using a GLM, we found that the percent area was the most significant parameter describing a threshold to soma activation. We then compared the delay of domain activation and its distance from the soma, and recreated previous results that suggest a centripetal model of astrocytic calcium responses from the distal arborizations to the soma (*Fedotova et al., 2023*; *Rupprecht et al., 2024*). Here, we went a step further and discovered that soma activation switches this directionality for astrocytic calcium surge to propagate outward in a centrifugal manner away from the soma. Taken together, these results demonstrate the integrative potential of astrocyte calcium responses and characterize further the astrocyte calcium surge to relay this other parts of the astrocyte.

We were able to discover this general phenomenon of astrocyte physiology through the use of a novel computational tool that allowed us to combine almost 1000 astrocyte responses. Variation is rife in biological systems, and there are sure to be eccentricities within astrocyte calcium responses. Here, we focused on grouped data to better understand what appears to be an intrinsic property of astrocyte physiology. We used different statistical examinations and tested our hypothesis in vivo and in situ, and all these methods together provide a more complete picture of the existence of a spatial threshold for astrocyte calcium surge.

The investigation of the spatial threshold could be improved in the future in a number of ways. One being the use of state-of-the-art imaging in 3D (*Bindocci et al., 2017*). While the original publication using 3D imaging to study astrocyte physiology does not necessarily imply that there would be different calcium dynamics in one axis over another, the three-dimensional examination of the spatial threshold could refine the findings we present here. To better control the system, mice imaged here were under anesthesia, and this is a method that has been used to characterize many foundational physiological results in the field (*Hubel and Wiesel, 1962*; *Mountcastle et al., 1957*). However, assessing the spatial threshold in awake freely moving animals would be the next logical step. In this study, we chose to limit our examinations of calcium activity that was within the bounds determined by SR101 staining. Much work has shown that astrocyte territories are more akin to sponge-like morphology with small microdomains making up the end feet of their arborizations (*Baldwin et al., 2024*). Here, we took a conservative approach to not incorporate these fine morphological processes and only take SR101-postive pixels for analysis in order to reduce the possible error of including a neighboring astrocyte or extracellular space in our analyses. Much work can be done to extend these results.

Detailed examinations of the subcellular roots of population activity is important in understanding network activity (*Buzsáki et al., 2012*; *Yuste, 2015*). Examining single-unit neuronal activity from a specific brain region and comparing to low-frequency recordings of the local field potential creates a reductionist description of network-mediated brain function (*Buzsáki and Draguhn, 2004*). The discovery and detailing of the action potential was landmark to understanding neuronal information processing capabilities as integrators at the single cell (*Yuste and Tank, 1996*). The determination of an astrocyte subcellular spatial threshold that underlies a calcium surge is an analogue to the action potential threshold found in neurons. Much like in neurons, the astrocyte spatial threshold is shown as a transformation of subcellular activity that underlies integrative properties.

IP$_3$R2-dependent calcium release has been shown to critically contribute to G-protein-coupled receptor-induced astrocyte calcium activity in the age range of mice imaged here (*Agulhon et al.,*

2008; *Kofuji and Araque, 2020*). Using *Itpr2*[-/-] mice, we found that IP$_3$R2 are required for somatic calcium responses and that they are necessary for the sensory-evoked responses to surpass the spatial threshold. Indeed, close examinations of astrocytic arborizations in *Itpr2*[-/-] mice showed domains still responded with calcium events, albeit at a reduced percentage compared to wildtype mice. Moreover, this reduction of domain activity in *Itpr2*[-/-] mice was steadily below the spatial threshold for calcium surge, suggesting a role for IP$_3$ in this physiological property. The subcellular calcium dynamics of astrocytes in *Itpr2*[-/-] mice have been shown previously (*Agarwal et al., 2017*; *Lines et al., 2020*; *Schmidt and Oheim, 2020*; *Srinivasan et al., 2015*; *Stobart et al., 2018*), yet present data further demonstrate that IP$_3$R2 are necessary for the propagation of astrocyte calcium surge. Outside of IP$_3$R2-mediated intracellular Ca$^{2+}$ increases, extracellular Ca$^{2+}$ entry into the cell has been shown (*Rungta et al., 2016*), and our study does not rule out this possibility.

Astrocyte calcium activity induces multiple downstream signaling cascades, such as the release of gliotransmitters (*Araque et al., 2014*; *de Ceglia et al., 2023*). Using patch-clamp recordings of a single nearby neuron we showed that a nearby population of astrocyte calcium surge is also correlated to the increase in SICs, previously demonstrated to be dependent on astrocytic vesicular release of glutamate (*Araque et al., 2000*; *Durkee et al., 2019b*; *Fellin et al., 2004*). The increase of SICs we observed from patching a single neuron is likely the integration of gliotransmitter release onto synapses from a group of nearby astrocytes. Indeed, subthreshold astrocyte calcium increases alone can trigger activity in contacted dendrites (*Di Castro et al., 2011*). An exciting avenue of future research would be to observe the impact of a single astrocyte calcium surge on nearby neurons (*Refaeli and Goshen, 2022*). How many neurons would be affected, and would this singular event be observable through patch clamp from a single neuron? The output of astrocyte calcium surge is equally important to network communication as the labeling of astrocyte calcium surge, as it identifies a biologically relevant effect onto nearby neurons. Many downstream signaling mechanisms may be activated following astrocyte calcium surge, and the effect of locally concentrated domain activity vs astrocyte calcium surge should be studied further on different astrocyte outputs.

In addition to normal brain function, many neurological disorders have been shown to have a cause at the cellular level that translate up to aberrant network brain function. Examples include: closer inspections of Alzheimer's disease have uncovered aberrant synaptic activity early on in the disease that may underly network dysfunction and cognitive processes (*Selkoe, 2002*), increased cellular excitability contributes to epileptic seizure activity (*Cohen et al., 2002*), and NMDA dysfunction in schizophrenia impairs long-range neuronal synchronization contributing to altered cognitive states (*Olney et al., 1999*). Closer examinations into altered subcellular astrocyte activity may also uncover contributions to neurological disorders. By understanding the root of the cause, novel translational diagnostics and therapeutics for brain disorders may be found.

Considering that a single astrocyte can contact ~100,000 synapses (*Bushong et al., 2002*) that can independently trigger the astrocyte calcium signal (*Covelo and Araque, 2018*; *Panatier et al., 2011*) and that can be independently regulated by gliotransmitters released through calcium-dependent mechanisms (*Araque et al., 2014*; *Savtchouk and Volterra, 2018*), the processes governing the intracellular expansion of the calcium signal may have relevant consequences on brain function by determining the spatial extension of astrocytic neuromodulation of synapses. In conclusion, by showing novel integrative properties of spatial information by astrocytes and the existence of a spatial threshold for the spread of the calcium signal and the subsequent gliotransmission, which is determined by astrocyte intrinsic properties, present findings identify novel physiological properties of astrocyte function that may add computational capabilities to brain information processing.

## Methods

### Key resources table

| Reagent type (species) or resource | Designation | Source or reference | Identifiers | Additional information |
|---|---|---|---|---|
| Strain, strain background (*Mus musculus*) | B6.Cg-Tg(Gfap-cre)77.6Mvs/2J | Jackson Laboratories | 024098 | Crossed with GCaMP6f mice to create Gfap-GCaMP6f transgenic mice. |

*Continued on next page*

*Continued*

| Reagent type (species) or resource | Designation | Source or reference | Identifiers | Additional information |
|---|---|---|---|---|
| Strain, strain background (*M. musculus*) | B6J.Cg-Gt(ROSA)26Sortm95.1 (CAG-GCaMP6f)Hze/MwarJ | Jackson Laboratories | 028865 | Crossed with GFAP-cre mice to create GFAP-GCaMP6f transgenic mice. |
| Strain, strain background (*M. musculus*) | Itpr2−/− mice | Ju Chen Kab | | |
| Chemical compound, drug | Urethane | Sigma-Aldrich | U2500 | |
| Chemical compound, drug | Sulforhodamine 101 | Sigma-Aldrich | S7635 | |
| Software, algorithm | MATLAB | Mathworks | RRID:SCR_001622 | |
| Recombinant DNA reagent | AAV5-GfaABC1d-22 GCaMP6f | UNC Vector Core | | |

## Proper animal use and care

All the procedures for handling and sacrificing animals were approved by the University of Minnesota Institutional Animal Care and Use Committee (IACUC) in compliance with the National Institutes of Health guidelines for the care and use of laboratory animals. We used both female and male transgenic animals that were 2–4 months of age, kept on a continuous 12 hr light/dark cycle and freely available to food and water. Expression of GCaMP6f in astrocytes was achieved by crossing *Gfap*-Cre mice with Gt(ROSA)26-lsl-GCaMP6f mice.

## Stereotaxic surgery for in vivo recordings

Mice were anesthetized with 1.8 mg/kg urethane administered intraperitoneally. Anesthetized mice were placed in a stereotaxic atop a heating pad controlled with an anal probe feedback to maintain body temperature (37°C), respiration was continuously monitored and faux tears were applied to prevent corneal dehydration. An incision was made down the midline of the scalp and the skin was parted to expose the skull. Screws were placed over the right frontal plate and interparietal plate. A craniotomy was made no more than 2 mm in diameter centered over the primary somatosensory cortex (S1; in mm from bregma: $-1_{a-p}$, $1.5_{m-l}$) (*Franklin, 2019*). After the dura was removed, SR101 was topically applied to the exposed cortex to label astrocytes (50 µM for 20 min) (*Rasmussen et al., 2016*). Agarose (1%) was made from artificial cerebrospinal fluid (ACSF) (containing in mM: NaCl 140, KCl 5, $MgCl_2$ 1, $CaCl_2$ 2, EDTA 1, HEPES-K 8.6, Glucose 10) and placed on the exposed cortex before fixing a glass coverslip over the craniotomy using dental cement. Finally, a frame was mounted onto the exposed skull using dental cement. In experiments testing $IP_3R2$ in calcium surge, 2 weeks before imaging mice were injected with adenovirus encoding GCaMP6f under the GfaABC1d promoter (AAV5-GfaABC1d-GCaMPf) into S1 (*Rupprecht et al., 2024*).

## In vivo two-photon calcium fluorescence imaging

In vivo imaging was performed in layers 2/3 (100–300 µm below the cortical surface) of the exposed mouse cortex with a Leica SP5 multiphoton upright microscope. Videos were obtained for 60 s over an area of 366×366 µm² at either 256×256 or 512×512 sized images with a sampling interval of 0.2–0.5 s. Red and green fluorescence was obtained in parallel to image calcium activity in identified SR101-labeled astrocytes.

## Peripheral stimulation

A bipolar electrode needle was placed in the hindpaw contralateral to the recorded cortical hemisphere. Square electrical pulses of 2 mA amplitude and 0.5 ms width were delivered at 2 Hz for 20 s. In experiments testing stimulation parameters were done to test different intensities (1, 2, 3 mA always with 2 Hz for 10 s) or variable frequencies (0.5, 1, 2, 5, 10 Hz at 2 mA for 10 s) or for different durations (1, 5, 10, 20 s at 2 mA and 2 Hz). Stimulus parameters were pseudorandomly ordered to differentially activate and characterize different levels of activation of astrocytes. Stimulations were separated by at least 2 min.

## Calcium image processing and analysis

All image processing and analysis was performed in the novel graphical user interface Calsee created in MATLAB (*Figure 1—figure supplement 1*; https://github.com/justinlines/Calsee; *Lines, 2020*). Previously published using in vivo and in situ data (*Baraibar et al., 2023*; *Corkrum and Araque, 2021*; *Lines et al., 2022*; *Lines et al., 2020*; *Nanclares et al., 2023*). Within Calsee, functional and structural video files can be loaded simultaneously. ROIs can be defined manually or automatically based on structural or functional imaging. In this study, structural images of SR101-stained astrocytes were used to manually create an outer border around individual astrocyte territories (*Bindocci et al., 2017*). These structurally defined ROIs labeling the outer boundary of individual astrocytes were refined using Calsee, which allows the user to click the center of the soma on a cell $(x_{soma}, y_{soma})$, marking the center of the cell in polar coordinates with radius $r_{cell}$ and angle $\theta_{cell}$ as defined in *Equations 1 and 2* (*Figure 1—figure supplement 1B*).

$$r_{cell} = \sqrt{\left(x - x_{soma}\right)^2 + \left(y - y_{soma}\right)^2} \tag{1}$$

$$\theta_{cell} = tan^{-1}\left(\frac{\left(y - y_{soma}\right)}{\left(x - x_{soma}\right)}\right) \tag{2}$$

To refine ROIs down to only SR101-positive pixels, we first define the soma. The fluorescence $F\left(r_j\right)$ of every concentric ring $j$ of radius $r_j$ is found as in *Equation 3* by averaging over $\theta$ and is used to plot the change in fluorescence vs radius (*Figure 1—figure supplement 1B and C*).

$$F\left(r_j\right) = \frac{1}{N}\sum_{i=0}^{2\pi} F\left(r_j, \theta_i\right) \tag{3}$$

The radius of the first ring from the center whose fluorescence falls below 50% of the center ring fluorescence $(F\left(r_1\right))$ is used as the radius of the soma $r_{soma}$. Arborizations are defined as regions within each ring that is 0.25 standard deviations above the median fluorescence of that ring out until the algorithm reaches the manually drawn cellular border. Next, astrocyte territories are further discretized into a grid of domains that are maximally sized $4.3 \times 4.3$ μm$^2$ square ROIs. At fine distal processes, ROIs were automatically reduced in size to only include SR101-positive pixels within the manually drawn territory boundary. These ROIs based on SR101 labeling were then used to quantify calcium activity from the simultaneously recorded green channel. The fluorescence traces were normalized by the average fluorescence of the 10 s preceding sensory stimulation onset. Event detection of calcium fluorescence was determined when the amplitude of the response was three times the standard deviation away from the average baseline amplitude.

Every calcium event following the delivery of stimulation in the domains of an astrocyte before its soma becomes active is referred as pre-soma events. Events happening in the domains after soma activation are referred as post-soma events. Accordingly, a cell whose soma becomes active at a given moment can be subdivided into pre-soma cell (all the activity of the cell prior to soma activation) and post-soma cell (all the activity of the cell following soma activation).

## Heaviside step function

The Heaviside step function below in *Equation 4* is used to mathematically model the transition from one state to the next and has been used in simple integrate and fire models (*Bueno-Orovio et al., 2008*; *Gerstner, 2000*).

$$\mathrm{H}(a) := \begin{cases} 0, a < a_T \\ 1, a \geq a_T \end{cases} \tag{4}$$

The Heaviside step function $H\left(a\right)$ is zero everywhere before the threshold area $(a_T)$ and one everywhere afterward. From the data shown in *Figure 4E* where each point $(S\left(a\right))$ is an individual astrocyte response with its percent area $(a)$ domains active and if the soma was active or not denoted by a 1 or 0, respectively. To determine $a_T$ in our data we iteratively subtracted $H\left(a\right)$ from $S\left(a\right)$ for all possible

values of $a_T$ to create an error term over $a$. The area of the minimum of that error term was denoted the threshold area.

### Slice experiments

Following rapid decapitation, brains were extracted and placed in a vibratome to create 350 µm thick brain slices that included the primary somatosensory cortex. Brain slices were left to incubate in ACSF containing (in mM): NaCl 124, KCl 2.69, $KH_2PO_4$ 1.25, $MgSO_4$ 2, $NaHCO_3$ 26, $CaCl_2$ 2, ascorbic acid 0.4, and glucose 10, and continuously bubbled with carbogen (95% $O_2$ and 5% $CO_2$) (pH 7.3). After incubation, brain slices were placed in a chamber with a perfusion system to image astrocytes as well as record neuronal membrane potential via patch clamp. To stimulate astrocytes locally, a pipette tip was lowered above the slice and used to apply a puff of 0.5 mM ATP. Like in vivo experiments, videos were obtained for 60 s with at least 2 min of interstimulus time. Processing and analysis were performed in the same manner as described above for in vivo experimental data.

### Statistical analysis

Astrocyte calcium quantifications were initially averaged, for data presented in *Figures 2, 4D, F, and 7E–G*, over all astrocytes of a single video and these values were used in statistical testing. All other data and statistical examination presented in *Figures 3, 4E, G–H, 5, 6, 7D* are based on individual astrocyte responses. Paired and unpaired two-tailed Student's t-tests were performed with $\alpha=0.05$ against the null hypothesis that no difference exists between the two groups. Correlations were confirmed using a Student's t-test against the null hypothesis that no correlation exist. To test the stimulus dependence of a stimulus-response curve, one-way ANOVAs were performed with $\alpha=0.05$ against the null hypothesis that no dependence exists. In the comparisons of two groups' response curves a two-way ANOVA was performed with $\alpha=0.05$ against the additional null hypotheses that the two groups are the same and no interaction exists. In some examinations following a significant ANOVA, multiple comparison testing was performed using Tukey's range test using $\alpha=0.05$ against the null hypothesis that no samples are different from each other. Sample sizes were based off of previous reports (*Lines et al., 2022*; *Lines et al., 2020*). Experimental ordering of stimulus intensity was randomized for each field of view. Only data from healthy preparations, for both in vivo and in situ experiments, were included. The GLM analysis was performed in MATLAB.

## Acknowledgements

We would like to thank Dana Deters for technical support; Johanna de la Cruz and Julio Esparza for MATLAB helpful advice; Michelle Corkrum, Caitlin Durkee, Ana Covelo, Mario Martin-Fernandez, and Austin Ferro for helpful suggestions; Mark Sanders, Guillermo Marques, and Jason Mitchell at the University of Minnesota – University Imaging Centers for assistance using the Leica SP5 multiphoton upright microscope; This work was supported by Ministry of Science and Innovation (#PID2021-122586NB-I00), (#RTI2018-094887-B-I00), and Fondo Europeo de Desarrollo Regional (FEDER) to MN; National Institutes of Health (NIMH R01MH119355 and NIDA R01DA048822) and Department of Defense (W911NF2110328) to AA; NIH-NIA (1F31AG057155-01A1) and University of Minnesota Doctoral Dissertation Fellowship to JL; Salvador de Madariaga Program (PRX19/00646) and Ministerio de Ciencia, Innovación y Universidades (BFU2017-88393-P), Spain, and AEI/FEDER, EU, to EDM; National Institutes of Health-MH (R01MH119355) to PK; Ministerio de Ciencia e Innovación (PID2019-105020GB-100), Spain, Ayudas para la Movilidad de Investigadores M-BAE (BA15/00078) del Instituto de Salud Carlos III, Spain, and co-funded by FEDER ('A way to make Europe') to JA.

## Additional information

### Funding

| Funder | Grant reference number | Author |
| --- | --- | --- |
| National Institute of Mental Health | R01MH119355 | Alfonso Araque Paulo Kofuji |

| Funder | Grant reference number | Author |
|---|---|---|
| National Institute on Drug Abuse | R01DA048822 | Alfonso Araque |
| Department of Defense Education Activity | W911NF2110328 | Alfonso Araque |
| Ministerio de Ciencia e Innovación | PID2021-122586NB-I00 | Marta Navarrete |
| Ministerio de Ciencia e Innovación | RTI2018-094887-B-I00 | Marta Navarrete |
| National Institute on Aging | F31AG057155 | Justin Lines |
| Ministerio de Ciencia, Innovación y Universidades | BFU2017-88393-P | Eduardo D Martin |
| Ministerio de Ciencia e Innovación | PID2019-105020GB-100 | Juan Aguilar |
| Fondo Europeo de Desarrollo Regional | | Marta Navarrete Eduardo D Martin |
| University of Minnesota | Doctoral Dissertation Fellowship | Justin Lines |
| Salvador de Madariaga Program | PRX19/00646 | Eduardo D Martin |
| Agencia Estatal de Investigación | | Eduardo D Martin |
| Instituto de Salud Carlos III | BA15/00078 | Juan Aguilar |

The funders had no role in study design, data collection and interpretation, or the decision to submit the work for publication.

## Author contributions

Justin Lines, Conceptualization, Data curation, Software, Formal analysis, Investigation, Visualization, Methodology, Writing – original draft, Writing – review and editing; Andres Baraibar, Formal analysis, Investigation, Methodology; Carmen Nanclares, Data curation, Validation, Investigation, Methodology; Eduardo D Martin, Conceptualization, Investigation, Methodology; Juan Aguilar, Conceptualization, Supervision, Investigation, Methodology, Writing – review and editing; Paulo Kofuji, Conceptualization, Resources, Supervision, Investigation, Writing – review and editing; Marta Navarrete, Conceptualization, Investigation, Writing – review and editing; Alfonso Araque, Conceptualization, Resources, Supervision, Funding acquisition, Investigation, Visualization, Writing – review and editing

## Author ORCIDs

Justin Lines ⬦ https://orcid.org/0000-0002-5316-3933
Juan Aguilar ⬦ https://orcid.org/0000-0002-8070-3923
Marta Navarrete ⬦ https://orcid.org/0000-0003-2097-4788
Alfonso Araque ⬦ https://orcid.org/0000-0003-3840-1144

## Ethics

Research involving vertebrate animals was done at the University of Minnesota following protocols reviewed and approved by the University of Minnesota Institutional Animal Care and Use Committee (UMN IACUC) number 2001-37757A. The animals were cared for by Veterinary Services under a currently AAALAC approved program. The animals were housed in NIH-approved facilities and are observed daily by technicians. Unusual events are reported to the on call veterinarian, as well as to the investigator according to posted protocols. Other maintenance veterinary care was conducted according to NIH guidelines on the Use and Care of Animals. Facilities were inspected regularly according to NIH and AAALAC guidelines.

Reviewer #2 (Public review): https://doi.org/10.7554/eLife.90046.3.sa1
Reviewer #3 (Public Review): https://doi.org/10.7554/eLife.90046.3.sa2

Author response https://doi.org/10.7554/eLife.90046.3.sa3

---

## Additional files

### Supplementary files
• MDAR checklist

### Data availability
The datasets generated during and/or analyzed during the current study are available at G-Node. The code used in the study can be found at Github (*Lines, 2020*).

The following dataset was generated:

| Author(s) | Year | Dataset title | Dataset URL | Database and Identifier |
|---|---|---|---|---|
| Justin L | 2024 | Lines_et_al_eLife | https://gin.g-node.org/justinlines/Lines_et_al_eLife | G-Node, justinlines/Lines_et_al_eLife |

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
