## [Editor Report · eLife Assessment]

This study presents **valuable** findings that add to our understanding of cortical astrocytes, which respond to synaptic activity with calcium release in subcellular domains that can proceed to larger calcium waves. The proposed concept of a spatial "threshold" is based on **solid** evidence from in vivo and ex vivo imaging data and the use of mutant mice. Details of the specific threshold must be taken with caution and are necessarily incomplete, but may be supported by additional experiments with higher resolution in space and time in the future.

---

## [Referee Report · Reviewer #2 (Public review)]

Summary

Lines et al investigate the integration of sensory-evoked calcium signals in astrocytes of the primary somatosensory cortex in anesthetized mice. More precisely, their goal is to better characterize the mechanisms that govern the emergence of whole-cell events in astrocytes, here referred to as calcium surges. As a single astrocyte communicates with hundreds of thousands of synapses simultaneously, understanding the spatial and temporal integration of calcium signals in astrocytes and the mechanisms governing these phenomena is of tremendous importance to deepen our understanding of signal processing in the central nervous system. In line with previous reports in the field, the authors find that most signals originate in the arborization of astrocytes, occasionally leading to somatic and whole-cell events. On average, the latter occur following domain activity closer to the soma, suggesting a centripetal propagation of signals leading to somatic events. Moreover, they observe that the distance from the soma to active domains increases with time after somatic events, suggesting a potential centrifugal propagation of signals post-somatic activity. The results suggest that most calcium surges depend on the expression of IP3R2, the main calcium channel in astrocytes, located at the membrane of the endoplasmic reticulum. Finally, they report a correlation between the percentage of active domains in the astrocyte "arbor", the emergence of a somatic event, and the frequency of slow inward currents from neighboring neurons. The main claim of this manuscript is that there would be a spatial threshold inherent to astrocytes of ~23% of domain activation above which a calcium surge is observed. Although the study provides data and concepts that are important for the glia field, the conclusions seem a little too assertive and general with respect to what can be deduced from the data and methods used.

Strengths

The major strength of this study is the experimental approach that allowed the authors to obtain numerous and informative calcium recordings in vivo in the somatosensory cortex in mice in response to sensory stimuli as well as in situ. Notably, they developed an interesting approach to modulate the percentage of active domains in the astrocyte arborization by varying the intensity of peripheral stimulation (its amplitude, frequency, or duration). The question investigated is important as the mechanisms governing signal integration in astrocytes and its effect on neighboring cells are poorly understood.

Weaknesses

The major weakness of the manuscript is the method used to analyze and quantify calcium activity, which mostly relies on the analysis of averaged data and overlooks the variability of the signals measured. As a result, the main claims from the manuscript seem to be incompletely supported by the data.

Although the revised version includes more discussion on the experiments that could be done to extend the results from this study, more discussion would be needed to clarify the limitations on what can be deduced from the proposed experimental and analytical design. Notably, the analysis pipeline seems biased by the assumption of the existence of a spatial threshold dictating the emergence of global calcium events in astrocytes. Although there is a clear linear correlation between the percentage of active somas and the percentage of active domains in the arborization (Figure 2 panel F), concluding on the existence of an inherent threshold of domain activity is not completely supported by the data (see e.g. Figure 2 panel F or Figure 4 panel E). It would probably be more accurate to report that most somatic events occur when the percentage of arbor domains being active is above 21-24% (95% confidence interval of the reported threshold). Thus, some of the conclusions from the manuscript, such as p.14 l.34-35 " spatial threshold of domains that needs to be reached in order to lead to soma activation", seem a bit too assertive as some astrocytes did display soma activation with a much smaller percentage of active domains or on the contrary, no somatic event despite domain activity way above the threshold. Similarly, as Figure 6 demonstrates a strong effect of IP3R2 knock-out on somatic activation but reports a non-zero probability of soma activity in IP3R2 -/- mice (panel F), the conclusion that IP3R2 are necessary to trigger an astrocytic calcium surge seems a bit too strong. Finally, the results reported in Figure 7 demonstrate the existence of a strong correlation between SICs, the percentage of active astrocyte domains on, and somatic activation, so that the conclusion "These results indicate that spatial threshold of the astrocyte calcium surge has a functional impact on gliotransmission" (l.4&-48 page 13) also seems a bit too assertive.

---

## [Referee Report · Reviewer #3 (Public Review)]

Summary:

The study aims to elucidate the spatial dynamics of subcellular astrocytic calcium signaling. Specifically, they elucidate how subdomain activity above a certain spatial threshold (~23% of domains being active) heralds a calcium surge that also affects the astrocytic soma. Moreover, they demonstrate that processes on average are included earlier than the soma and that IP3R2 is necessary for calcium surges to occur. Finally, they associate calcium surges with slow inward currents.

The revised manuscript is improved compared to the first iteration. While some concerns have been addressed, my main critique pertaining to ROI approach/sampled area, statistical analyses and anesthesia are in my view still important caveats of the study that I think should have been even more clearly addressed in the manuscript.

Strengths:

The study addresses an interesting topic that is only partially understood. The study uses multiple methods including in vivo two-photon microscopy, acute brain slices, electrophysiology, pharmacology, and knockout models. The conclusions are strengthened by the same findings in both in vivo anesthetized mice and in brain slices.

Weaknesses:

The method that has been used to quantify astrocytic calcium signals only analyzes what seems to be a small proportion of the total astrocytic domain on the example micrographs, where a structure is visible in the SR101 channel (see for instance Reeves et al. J. Neurosci. 2011, demonstrating to what extent SR101 outlines an astrocyte). This would potentially heavily bias the results: from the example illustrations presented it is clear that the calcium increases in what is putatively the same astrocyte goes well beyond what is outlined with automatically placed small ROIs. The smallest astrocytic processes are an order of magnitude smaller than the resolution of optical imaging and would not be outlined by either SR101 or with the segmentation method judged by the ROIs presented in the figures. Completely ignoring these very large parts of the spatial domain of an astrocyte, in particular when making claims about a spatial threshold, seems inappropriate. Several recent methods published use pixel-by-pixel event-based approaches to define calcium signals. The data should have been analyzed using such a method within a complete astrocyte spatial domain in addition to the analyses presented. Also, the authors do not discuss how two-dimensional sampling of calcium signals from an astrocyte that has processes in three dimensions (see Bindocci et al, Science 2017) may affect the results: if subdomain activation is not homogeneously distributed in the three-dimensional space within the astrocyte territory, the assumptions and findings between a correlation between subdomain activation and somatic activation may be affected.

Authors reply: In order to reduce noise from individual pixels, we chose to segment astrocyte arborizations into domains of several pixels. As pointed out previously, including pixels outside of the SR101-positive territory runs the risk of including a pixel that may be from a neighboring cell or mostly comprised of extracellular space, and we chose the conservative approach to avoid this source of error. We agree that the results have limitations from being acquired in 2D instead of 3D, but it is likely to assume the 3D astrocyte is homogeneously distributed and that the 2D plane is representative of the whole astrocyte. Indeed, no dimensional effects were reported in Bindocci et al, Science 2017. We have included a paragraph in the discussion to address this limitation in our study on P15, L23-27:

"The investigation of the spatial threshold could be improved in the future in a number of ways. One being the use of state-of-the-art imaging in 3D(Bindocci et al., 2017). While the original publication using 3D imaging to study astrocyte physiology does not necessarily imply that there would be different calcium dynamics in one axis over another, the three-dimensional examination of the spatial threshold could refine the findings we present here.

Comments on revisions: It is good that 3D imaging aspects are mentioned as a limitation, and I agree that Bindocci et al. do not necessarily suggest that results in this manuscript would have been different if also the third spatial dimension was included in the analyses. However, the way I see it, the added analyses and text changes throughtout still do not adequately address my concern pertaining to basing a spatial threshold on a fraction of the astrocyte territory.

The study uses a heaviside step function to define a spatial 'threshold' for somata either being included or not in a calcium signal. However, Fig 4E and 5D showing how the method separates the signal provide little understanding for the reader. The most informative figure that could support the main finding of the study, namely a ~23% spatial threshold for astrocyte calcium surges reaching the soma, is Fig. 4G, showing the relationship between the percentage of arborizations active and the soma calcium signal. A similar plot should have been presented in Fig 5 as well. Looking at this distribution, though, it is not clear why ~23% would be a clear threshold to separate soma involvement, one can only speculate how the threshold for a soma event would influence this number. Even if the analyses in Fig. 4H and the fact that the same threshold appears in two experimental paradigms strengthen the case, the results would have been more convincing if several types of statistical modeling describing the continuous distribution of values presented in Fig. 4E (in addition to the heaviside step function) were presented.

Authors reply: We agree with the reviewer and have added to the paper a discussion for our justification on the use of the Heaviside step function, and have included this in the methods section. We chose the Heaviside step function to represent the on/off situation that we observed in the data that suggested a threshold in the biology. We agree with the reviewer that Fig. 4G is informative and demonstrates that under 23% most of the soma fluorescence values are clustered at baseline. We agree that a different statistical model describing the data would be more convincing and confirmed the spatial threshold with the use of a confidence interval in the text and supported the use of percent domains active for this threshold over other properties such as spatial or temporal clustering using a general linear model. P18-19, L34-2:

"Heaviside step function

The Heaviside step function below in equation 4 is used to mathematically model the transition from one state to the next and has been used in simple integrate and fire models (Bueno-Orovio et al., 2008; Gerstner, 2000).(4)H(a):={0,a<aT1,a≥aT

The Heaviside step function 𝐻(𝑎) is zero everywhere before the threshold area (𝑎T) and one everywhere afterwards. From the data shown in Figure 4E where each point (𝑆(𝑎)) is an individual astrocyte response with its percent area (𝑎) domains active and if the soma was active or not denoted by a 1 or 0 respectively. To determine 𝑎T in our data we iteratively subtracted 𝐻(𝑎) from 𝑆(𝑎) for all possible values of 𝑎T to create an error term over 𝑎. The area of the minimum of that error term was denoted the threshold area.

Comments on revisions: Even with the added explanations, I am still not sure that the data show a specific threshold, or that the statistical model enforce a threshold onto the data. The data in Fig. 4G does not in my view clearly show a clear threshold as suggested. The analyses are strengthened with an added statistical modeling, however, the details of the modeling is not presented in the manuscript as far as I can see. As a bare minimum the statistical packages/tools used, the model details and goodness of fit as residual plots must be shown/commented.

The description of methods should have been considerably more thorough throughout. For instance which temperature the acute slice experiments were performed at, and whether slices were prepared in ice-cold solution, are crucial to know as these parameters heavily influence both astrocyte morphology and signaling. Moreover, no monitoring of physiological parameters (oxygen level, CO2, arterial blood gas analyses, temperature etc) of the in vivo anesthetized mice is mentioned. These aspects are critical to control for when working with acute in vivo two-photon microscopy of mice; the physiological parameters rapidly decay within a few hours with anesthesia and following surgery.

Authors reply: We have increased the thoroughness of our methods section. Especially including that body temperature and respiration were indeed monitored throughout anesthesia.

Comments on revisions: Bath temperature for slice experiments, or cutting conditions are still not reported. For the in vivo experiments, it must be commented that this level of physiological monitoring for acute in vivo brain physiology experiments (self breathing, no control of O2/CO2) is barely adequate and could represent a considerable caveat of the study.

---

## [Author Response]

The following is the authors’ response to the original reviews.

eLife AssessmentBuilding on their own prior work, the authors present valuable findings that add to our understanding of cortical astrocytes, which respond to synaptic activity with calcium release in subcellular domains that can proceed to larger calcium waves. The proposed concept of a spatial "threshold" is based on solid evidence from in vivo and ex vivo imaging data and the use of mutant mice. However, details of the specific threshold should be taken with caution and appear incomplete unless supported by additional experiments with higher resolution in space and time.

We thank the reviewers and editors for the positive assessment of our work as containing valuable findings that add to our understanding of cortical astrocytes. We also appreciate their positive appraisal of the proposed concept of a spatial threshold supported by solid evidence.

Regarding their specific comments, we truly appreciate them because they have helped to clarify issues and to improve the study. Point-by-point responses to these comments are provided below. Regarding the general comment on the spatial and temporal resolution of our study, we would like to clarify that the spatial and temporal resolution used in the current study (i.e., 2 - 5 Hz framerate using a 25x objective with 1.7x digital zoom with pixels on the order of 1 µm2) is within the norm in the field, does not compromise the results, nor diminish the main conceptual advancement of the study, namely the existence of a spatial threshold for astrocyte calcium surge.

We respect the thoughtfulness of the reviewers and editors towards improving the paper.

**Public Reviews:**

**Reviewer #1 (Public Review):**
Lines et al., provide evidence for a sequence of events in vivo in adult anesthetized mice that begin with a footshock driving activation of neural projections into layer 2/3 somatosensory cortex, which in turn triggers a rise in calcium in astrocytes within "domains" of their "arbor". The authors segment the astrocyte morphology based on SR101 signal and show that the timing of "arbor" Ca2+ activation precedes somatic activation and that somatic activation only occurs if at least {greater than or equal to}22.6% of the total segmented astrocyte "arbor" area is active. Thus, the authors frame this {greater than or equal to}22.6% activation as a spatial property (spatial threshold) with certain temporal characteristics - i.e., must occur before soma and global activation. The authors then elaborate on this spatial threshold by providing evidence for its intrinsic nature - is not set by the level of neuronal stimulus and is dependent on whether IP3R2, which drives Ca2+ release from the endoplasmic reticulum (ER) in astrocytes, is expressed. Lastly, the authors suggest a potential physiologic role for this spatial threshold by showing ex vivo how exogenous activation of layer 2/3 astrocytes by ATP application can gate glutamate gliotransmission to layer 2/3 cortical neurons - with a strong correlation between the number of active astrocyte Ca2+ domains and the slow inward current (SIC) frequency recorded from nearby neurons as a readout of glutamatergic gliotransmission. This is interesting and would potentially be of great interest to readers within and outside the glia research community, especially in how the authors have tried to systematically deconstruct some of the steps underlying signal integration and propagation in astrocytes. Many of the conclusions posited by the authors are potentially important but we think their approach needs experimental/analytical refinement and elaboration.

We thank the reviewer for her/his positive appraisal and comments that has helped us to improve the study. In response to their insights, we aim to address the key points raised below:

(1) Sequence of Events: We acknowledge the reviewer's interest in our findings regarding the sequence of events. We have provided a more detailed description of the methods and results to clarify the spatiotemporal relationships between domain activation and spatiotemporal clustering, to centripetal and centrifugal calcium propagation in relation to soma activation.

(2) Spatial Threshold: The reviewer accurately identifies our characterization of a spatial threshold (≥22.6% activation) with temporal characteristics as a crucial aspect of our study. We have expanded upon this concept by offering a clearer illustration of how this threshold relates to somatic and global activation.

(3) Intrinsic Nature of Spatial Threshold: The reviewer's insightful observation regarding the inherent quality of the spatial threshold, regardless of its dependence on neuronal stimuli is noteworthy. We have provided additional details to substantiate this claim, shedding more light on the fundamental nature of this phenomenon.

(4) Physiological Implications: The reviewer rightly highlights the potential physiological significance of our findings, particularly in relation to gliotransmission in cortical neurons. We have enhanced our discussion by elaborating on the implications of these observations.

The primary issue for us, and which we would encourage the authors to address, relates to the low spatialtemporal resolution of their approach. This issue does not necessarily compromise the concept of a spatial threshold, but more refined observations and analyses are likely to provide more reliable quantitative parameters and a more comprehensive view of the mode of Ca2+ signal integration in astrocytes.

We agree with the reviewer that our spatial-temporal resolution (2 – 5 Hz framerate using a 25x objective and 1.7x digital zoom with pixels on the order of 1 µm) does not compromise the proposed concept of the existence of a spatial threshold for the intracellular calcium expansion.

For this reason, and because their observations might be perceived as both a conceptual and numerical standard in the field, we believe that the authors should proceed with both experimental and analytical refinement. Notably, we have difficulty with the reported mean delays of astrocyte Ca2+ elevations upon sensory stimulation. The 11s delay for response onset in "arbor" and 13s in the soma are extremely long, and we do not think they represent a true physiologic latency for astrocyte responses to the sensory activity. Indeed, such delays appear to be slower even than those reported in the initial studies of sensory stimulation in anesthetized mice with limited spatial-temporal resolution (Wang et al. Nat Neurosci., 2006) - not to say of more recent and refined ones in awake mice (Stobart et al. Neuron, 2018) that identified even sub-second astrocyte Ca2+ responses, largely preserved in IP3R2KO mice. Thus, we are inclined to believe that the slowness of responses reported here is an indicator of experimental/analytical issues. There can be several explanations of such slowness that the authors may want to consider for improving their approach: (a) The authors apparently use low zoom imaging for acquiring signals from several astrocytes present in the FOV: do all of these astrocytes respond homogeneously in terms of delay from sensory stimulus? Perhaps some are faster responders than others and only this population is directly activated by the stimulus. Others could be slower in activation because they respond secondarily to stimuli. In this case, the authors could focus their analysis specifically on the "fast-responding population". (b) By focusing on individual astrocytes and using higher zoom, the authors could unmask more subtle Ca2+ elevations that precede those reported in the current manuscript. These signals have been reported to occur mainly in regions of the astrocyte that are GCaMP6-positive but SR101-negative and constitute a large percentage of its volume (Bindocci et al., 2017). By restricting analysis to the SR101-positive part of the astrocyte, the authors might miss the fastest components of the astrocyte Ca2+ response likely representing the primary signals triggered by synaptic activity. It would be important if they could identify such signals in their records, and establish if none/few/many of them propagate to the SR-101-positive part of the astrocyte. In other words, if there is only a single spatial threshold, the one the authors reported, or two or more of them along the path of signal propagation towards the cell soma that leads eventually to the transformation of the signal into a global astrocyte Ca2+ surge.

We thank the reviewer for these excellent and important comments. The qualm with the mean delays of astrocyte activation is indeed a result of averaging together astrocyte responses to a 20 second stimulus. Indeed, astrocyte responses are heterogeneous and many astrocytes respond much quicker, as can be seen in example traces in Figs. 1D, 1G, and 3C. Indeed, with any biological system variability exists, however here we take the averaged responses in order to identify a general property of astrocyte calcium dynamics: the existence of the concept of a spatial threshold for astrocyte calcium surge. We have now included a paragraph in the Discussion section on this subject on P15, L16-22:

“We were able to discover this general phenomenon of astrocyte physiology through the use of a novel computational tool that allowed us to combine almost 1000 astrocyte responses. Variation is rife in biological systems, and there are sure to be eccentricities within astrocyte calcium responses. Here, we focused on grouped data to better understand what appears to be an intrinsic property of astrocyte physiology. We used different statistical examinations and tested our hypothesis in vivo and in situ, and all these methods together provide a more complete picture of the existence of a spatial threshold for astrocyte calcium surge.“

The specialized work of Stobart et al. 2018, was focused more on the fast activation of microdomain subpopulations than the induction of later somatic activation. Indeed, Stobart et al. 2018 and Wang et al. 2006 also found that somatic responses of astrocytes were delayed in the range of seconds. Importantly, Wang et al., 2006 describe that the activation of astrocytes is frequency dependent, that is, the higher the frequency, the faster and higher the activation. In the present, work we stimulated at just 2 Hz to better investigate the spatial threshold. Excitingly, the results showed by Stobart et al., 2018 agree with ours, Rupprecht et al. 2024 and Fedotova et al. 2023, that there is a sequence of activation from the domains to the somas, which could be due to the time that is required for the summation of the initial microdomain signal to reach a threshold capable to activate the soma. These above referenced studies have many similarities with our own but are different in the underlying scientific question that led to diverging methodology, however we want to stress that we agree with the reviewers that our methods provide sufficient evidence for the cell-scale scientific phenomenon that we are studying, which is the spatial threshold for astrocyte calcium surge. Finally, we have included an additional figure (new Figure 5) that only looks at the calcium dynamics of early responding cells and found no significant difference in the spatial threshold in this population compared to our original quantification.

In this context, there is another concept that we encourage the authors to better clarify: whether the spatial threshold that they describe is constituted by the enlargement of a continuous wavefront of Ca2+ elevation, e.g. in a single process, that eventually reaches 22.6% of the segmented astrocyte, or can it also beconstituted by several distinct Ca2+ elevations occurring in separate domains of the arbor, but overall totaling 22.6% of the segmented surface? Mechanistically, the latter would suggest the presence of a general excitability threshold of the astrocyte, whereas the former would identify a driving force threshold for the centripetal wavefront. In light of the above points, we think the authors should use caution in presenting and interpreting the experiments in which they use SIC as a readout. Their results might lead some readers to bluntly interpret the 22.6% spatial threshold as the threshold required for the astrocyte to evoke gliotransmitter release. Indeed, SIC are robust signals recorded somatically from a single neuron and likely integrate activation of many synapses all belonging to that neuron. On the other hand, an astrocyte impinges in a myriad of synapses belonging to several distinct neurons. In our opinion, it is quite possible that more local gliotransmission occurs at lower Ca2+ signal thresholds (see above) that may not be efficiently detected by using SIC as a readout; a more sensitive approach, such as the use of a gliotransmitter sensor expressed all along the astrocyte plasma-membrane could be tested to this aim.

The reviewer raised an excellent point. Whether the spatial threshold of 22.6% occur in the segmented astrocyte or may be reached occurring in separate domains of the arbor, is an important question and we address this by the inclusion of a novel analysis shown in the new figure (new Figure 5) in the revised version of the manuscript. In this new analysis, we demonstrate that the average distance between domain activation is not significantly different between subthreshold activity and the activity that precedes or follows the suprathreshold cellular activation. In contrast, we do find a significant difference in the average time between domain activation between subthreshold activity and activity that precedes and follows suprathreshold activation. We go further with a generalized linear model to show that percent area of active domains and temporal clustering is related to soma activation and not spatial clustering. This suggests that domain activation doesn’t need to be spatially clustered together to induce soma activation and subsequent calcium surge, but more importantly, domain activation must be over the spatial threshold and occur within a timeframe. This has been added to the Results on P10, L2-40:

“Our results demonstrate the relationship between the percentage of active domains and soma activation and subsequent calcium surge. Next, we were interested in the spatiotemporal properties of domain activity leading up to and during calcium surge. Because we imaged groups of astrocytes, we were able to constrain our analyses to fast responders (onset < median population onset) in order to evaluate astrocytes that were more likely to respond to neuronal-evoked sensory stimulation and not nearby astrocyte activation (Figure 5A). In this population the spatial threshold was 23.8% within the 95% confidence intervals of [21.2%, 24.0%]. First, we created temporal maps, where each domain is labeled as its onset relative to soma activation, of individual astrocyte calcium responses to study the spatiotemporal profile of astrocyte calcium surge (Bindocci et al., 2017; Rupprecht et al., 2024) (Figure 5B). Using temporal maps, we quantified the spatial clustering of responding domains by measuring the average distance between active domains. We found that the average distance between active domains in subthreshold astrocyte responses were not significantly different from pre-soma suprathreshold activity (16.3 ± 0.4 µm in No-soma cells versus 16.2 ± 0.3 µm in Pre-soma cells, p = 0.75; n = 286 No-soma vs n = 326 Pre-soma, 30 populations and 3 animals; Figure 5C). Following soma activation, astrocyte calcium surge was marked with no significant change in the average distance between active domains (16.0 ± 0.3 µm in Post-soma cells versus 16.3 ± 0.4 µm in No-soma cells, p = 0.57 and 16.2 ± 0.3 µm in Presoma cells, p = 0.31; n = 326 soma active and n = 286 no soma active, 30 populations and 3 animals; Figure 5C). Taken together this suggests that on average domain activation happens in a nonlocal fashion that may illustrate the underlying nonlocal activation of nearby synaptic activity. Next, we interrogated the temporal patterning of domain activation by quantifying the average time between domain responses, and found that the average time between domain responses was significantly decreased in pre-soma suprathreshold activity compared to subthreshold activities without subsequent soma activation (9.4 ± 0.3 s in No-soma cells versus 4.4 ± 0.2 s in Pre-soma cells, p < 0.001; n = 326 soma active vs n = 286 not soma active, 30 populations and 3 animals; Figure 5D). The average time between domain activation was even less after the soma became active during calcium surge (2.1 ± 0.1 s in Post-soma versus 9.4 ± 0.3 s in No-Soma cells, p < 0.001 and 4.4 ± 0.1 s in Pre-soma cells, p < 0.001; n = 326 soma active and n = 286 not soma active, 30 populations and 3 animals; Figure 5D). This corroborates our findings in Figure S2 and highlights the difference in temporal profiles between subthreshold activity and astrocyte calcium surge.

We then tested the contribution of each of our three variables describing domain activation (percent area, average distance and time) to elicit soma activation by creating a general linear model. We found that overall, there was a significant relationship between these variables and the soma response (p = 5.5e-114), with the percent area having the largest effect (p = 3.5e-70) followed by the average time (p = 3.6e-7), and average distance having no significant effect (p = 0.12). Taken together this suggests that the overall spatial clustering of active domains has no effect on soma activation, and the percent area of active domains within a constrained time window having the largest effect.”

Regarding comments on SIC, we fully agree with the reviewer. In the revised version of the manuscript, we have included text in the discussion to ensure the correct interpretation of the results, i.e., the observed 22.6% spatial threshold for the SIC does not necessarily indicate an intrinsic property of gliotransmitter release; rather, since SICs have been shown to be calcium-dependent, it is not surprising that their presence, monitored at the whole-cell soma, matches the threshold for the intracellular calcium extension. We have added to the Discussion P16, L15-30:

“Astrocyte calcium activity induces multiple downstream signaling cascades, such as the release of gliotransmitters (Araque et al., 2014; de Ceglia et al., 2023). Using patch-clamp recordings of a single nearby neuron we showed that a nearby population of astrocyte calcium surge is also correlated to the increase in slow inward currents (SICs), previously demonstrated to be dependent on astrocytic vesicular release of glutamate (Araque et al., 2000; Durkee et al., 2019; Fellin et al., 2004). The increase of SICs we observed from patching a single neuron is likely the integration of gliotransmitter release onto synapses from a group of nearby astrocytes. Indeed, subthreshold astrocyte calcium increases alone can trigger activity in contacted dendrites (Di Castro et al., 2011). An exciting avenue of future research would be to observe the impact of a single astrocyte calcium surge on nearby neurons (Refaeli and Goshen, 2022). How many neurons would be affected, and would this singular event be observable through patch clamp from a single neuron? The output of astrocyte calcium surge is equally important to network communication as the labeling of astrocyte calcium surge, as it identifies a biologically relevant effect onto nearby neurons. Many downstream signaling mechanisms may be activated following astrocyte calcium surge, and the effect of locally concentrated domain activity vs astrocyte calcium surge should be studied further on different astrocyte outputs.”

Additional considerations are that the authors propose an event sequence as follows: stimulus - synaptic drive to L2/3 - arbor activation - spatial threshold - soma activation - post soma activation - gliotransmission. This seems reminiscent of the sequence underlying neuronal spike propagation - from dendrite to soma to axon, and the resulting vesicular release. However, there is no consensus within the glial field about an analogous framework for astrocytes. Thus, "arbor activation", "soma activation", and "post soma activation" are not established `terms-of-art´. Similarly, the way the authors use the term "domain" contrasts with how others have (Agarwal et al., 2017; Shigetomi et al., 2013; Di Castro et al., 2011; Grosche et al., 1999) and may produce some confusion. The authors could adopt a more flexible nomenclature or clarify that their terms do not have a defined structural-functional basis, being just constructs that they justifiably adapted to deal with the spatial complexity of astrocytes in line with their past studies (Lines et al., 2020; Lines et al., 2021).

We agree there is no consensus within the glial field about this event sequence. One major difference between this sequence of events and neuronal spike propagation is directionality from dendrite to soma to axon. It is unknown whether directionality of the calcium signal exists in astrocytes. However, our finding in Figure 5E suggests a directionality of centripetal propagation from the arborization to the soma to elicit calcium surge that leads to centrifugal propagation. In the Results on P10-11, L41-8:

“Recent work studying astrocyte integration has suggested a centripetal model of astrocyte calcium, where more distal regions of the astrocyte arborization become active initially and activation flows towards the soma (Fedotova et al., 2023; Rupprecht et al., 2024). Here, we confirm this finding, where activated domains located distal from the soma respond sooner than domains more proximal to the soma (linear correlation: p < 0.05, R2 = 0.67; n = 30 populations, 3 animals; Figure 4E). Next, we build upon this result to also demonstrate that following soma activation, astrocyte calcium surge propagates outward in a centrifugal pattern, where domains proximal to the soma become active prior to distal domains (linear correlation: p < 0.01, R2 = 0.89; n = 30 populations, 3 animals; Figure 4E). Together these results detail that intracellular astrocyte calcium follows a centripetal model until the soma is activated leading to a calcium surge that flows centrifugally. This suggests that astrocytes have the capabilities to integrate the nearby local synaptic population, and if this activity exceeds the spatial threshold then it leads to a whole-cell response that spreads outward.”

And in the Discussion P15, L3-15:

“Close examinations of the calcium surge uncovered distinct propagations whether before or after soma activation. Firstly, our analysis found that temporal clustering changed before and after calcium surge, with both being above subthreshold activity, and that this characteristic was absent when assessing spatial clustering. When comparing the percent area, spatial and temporal clustering of active domains using a GLM, we found that the percent area was the most significant parameter describing a threshold to soma activation. We then compared the delay of domain activation and its distance from the soma, and recreated previous results that suggest a centripetal model of astrocytic calcium responses from the distal arborizations to the soma (Fedotova et al., 2023; Rupprecht et al., 2023). Here, we went a step further and discovered that soma activation switches this directionality for astrocytic calcium surge to propagate outward in a centrifugal manner away from the soma. Taken together, these results demonstrate the integrative potential of astrocyte calcium responses and characterize further the astrocyte calcium surge to relay this other parts of the astrocyte.”

The term “microdomain” is used in the references above to define distal subcellular domains in contact with synapses, and in order to dissociate from this term we adopt the nomenclature “domain” to define all subcellular domains in the astrocyte arborization. These items have been discussed and clarified in the revised version of the manuscript on P5, L17-19:

“The concept of domain to define all subcellular domains in the astrocyte arborization should not be confused with the concept of microdomain, that usually refers to the distal subcellular domains in contact with synapses.”

Our previous points suggest that the paper would be significantly strengthened by new experimental observations focusing on single astrocytes and using acquisitions at higher spatial and temporal resolution. If the authors will not pursue this option, we encourage them to at least improve their analysis, and at the same time recognize in the text some limitations of their experimental approach as discussed above. We indicate here several levels of possible analytical refinement.

We believe our spatial (25x objective and 1.7x digital zoom with pixels on the order of 1µm) and temporal (2 – 5 Hz framerate) resolution is within the range used in the glial field. In any case the existence of a spatial threshold for astrocyte calcium surge is not compromised with the use of this imaging resolution.

The first relates to the selection of astrocytes being analyzed, and the need to focus on a much narrower subpopulation than (for example) 987 astrocytes used for the core data. This selection would take into greater consideration the aspects of structure and latency. With the structural and latency-based criteria for selection, the number of astrocytes to analyze might be reduced by 10-fold or more, making our second analytical recommendation much more feasible.

We agree that individual differences exist, however, establishing a general concept requires the sampling of many astrocytes. Nevertheless, we have included a new figure (new Figure 5) that analyzes early responders.

For structure-based selection - Genetically-encoded Ca2+ indicators such as GCaMP6 are in principle expressed throughout an astrocyte, even in regions that are not labelled by SR101. Moreover, astrocytes form independent 3D territories, so one can safely assume that the GCaMP6 signal within an astrocyte volume belongs to that specific astrocyte (this is particularly evident if the neighboring astrocytes are GCaMP6negative). Therefore, authors could extend their analysis of Ca2+ signals in individual astrocytes to the regions that are SR101-negative and try to better integrate fast signals in their spatial threshold concept. Even if they decided to be conservative on their methods, and stick to the astrocyte segmentation based on the SR-101 signal, they should acknowledge that SR101 dye staining quality can vary considerably between individual astrocytes within a FOV - some astrocytes will have much greater structural visibility in the distal processes than others. This means that some astrocytes may have segmented domains extending more distally than others and we think that authors should privilege such astrocytes for analysis. However, cases like the representative astrocytes shown in Figure 4A or Figure S1B, have segmented domains localized only to proximal processes near the soma. Accordingly, given the reported timing differences between "arbor" and "soma" activation, one might expect there to be comparable timing differences between domains that are distal vs proximal to the soma as well. Fast signals in peripheral regions of astrocytes in contact with synapses are largely IP3R2-independent (Stobart et al., 2018). However, the quality of SR101 staining has implications for interpreting the IP3R2 KO data. There is evidence IP3R2 KO may preferentially impact activity near the soma (Srinivasan et al., 2015). Thus, astrocytes with insufficient staining - visible only in the soma and proximal domains - might show a biased effect for IP3R2 KO. While not necessarily disrupting the core conclusions made by the authors based on their analysis of SR101-segmented astrocytes, we think results would be strengthened if astrocytes with sufficient SR101 staining - i.e. more consistent with previous reports of L2/3 astrocyte area (Lanjakornsiripan et al., 2018) - were only included. This could be achieved by using max or cumulative projections of individual astrocytes in combination with SR101 staining to construct more holistic structural maps (Bindocci et al., 2017).

We agree with the ideas concerning SR101, and indeed there could be variability in the origins of the astrocyte calcium signal. Astrocyte territory boundaries can be difficult to discern when both astrocytes express GCaMP6. Also, SR101-negative domains could encapsulate an area that is only partially that of astrocyte territory, including also extracellular space. Here we take a conservative approach to constrain ROIs to SR101positive astrocyte territory outlines without invading neighboring cells or extracellular space in order to reduce error in the estimate of a spatial threshold. The effect of IP3R2 KO preferentially impacting activity near the soma is interesting, and in line with our conclusions. We agree that the findings from SR101-negative pixels would not necessarily disrupt the core conclusions of the study, and the additional analysis suggested would further strengthen results. We have since included on the limitations of the study in the Discussion P15, L3137:

“In this study, we chose to limit our examinations of calcium activity that was within the bounds determined by SR101 staining. Much work has shown that astrocyte territories are more akin to sponge-like morphology with small microdomains making up the end feet of their distal arborizations (Baldwin et al., 2024). Here, we took a conservative approach to not incorporate these fine morphological processes and only take SR101-postive pixels for analysis in order to reduce the possible error of including a neighboring astrocyte or extracellular space in our analyses. Much work can be done to extend these results.”

For latency-based selection - The authors record calcium activity within a FOV containing at least 20+ astrocytes over a period of 60s, during which a 2Hz hindpaw stimulation at 2mA is applied for 20s. As discussed above, presumably some astrocytes in a FOV are the first to respond to the stimulus series, while others likely respond with longer latency to the stimulus. For the shorter-latency responders <3s, it is easier to attribute their calcium increases as "following the sensory information" projecting to L2/3. In other cases, when "arbor" responses occur at 10s or later, only after 20 stimulus events (at 2Hz), it is likely they are being activated by a more complex and recurrent circuit containing several rounds of neuron-glia crosstalk etc., which would be mechanistically distinct from astrocytes responding earlier. We suggest that authors focus more on the shorter latency response astrocytes, as they are more likely to have activity corresponding to the stimulus itself.

We agree that different times of astrocyte calcium increases may be due to different mechanisms outside of the astrocyte. We believe the spatial threshold will be intrinsic to these external variables; yet we believe that longer latency responses are physiological and may carry important information to determining the astrocyte calcium responses. Indeed, we have performed the spatial threshold analysis on early responders (first half of responding cells), and found the spatial threshold in that population (23.8%) is within the 95% confidence interval [21.2%, 24.0%]. Additionally, the slow responders were also within the confidence interval (22.6%).

The second level of analysis refinement we suggest relates specifically to the issue of propagation and timing for the activity within "arbor", "soma" and "post-soma". Currently, the authors use an ROI-based approach that segments the "arbor" into domains. We suggest that this approach could be supplemented by a more robust temporal analysis. This could for example involve starting with temporal maps that take pixels above a certain amplitude and plot their timing relative to the stimulus-onset, or (better) the first active pixel of the astrocyte. This type of approach has become increasingly used (Bindocci et al., 2017; Wang et al., 2019; Ruprecht et al., 2022) and we think its use can greatly help clarify both the proposed sequence and better characterize the spatial threshold. We think this analysis should specifically address several important points:

We agree that the creation of temporal maps from our own data would be interesting, and we provide the results of the suggested analysis within the new figure (new Figure 5) in the revised version of the manuscript. In this analysis we show that subthreshold, pre-soma and post-soma dynamics are significantly different in time. These added results of including temporal maps strengthen our claim of a spatial threshold, by quantifying the distinct temporal and spatial dynamics of domain activation before and after the spatial threshold is met (i.e. soma activation), and highlights differences in subthreshold and suprathreshold activity.

(1) Where/when does the astrocyte activation begin? Understanding the beginning is very important, particularly because another potential spatial threshold - preceding the one the authors describe in the paper - could gate the initial activation of more distal processes, as discussed above. This sequentially earlier spatial threshold could (for example) rely on microdomain interaction with synaptic elements and (in contrast) be IP3R2 independent (Srinivasan et al., 2015, Stobart et al., 2018). We would be interested to know whether, in a subset of astrocytes that meet the structure and latency criteria proposed above and can produce global activation, there is an initial local GCaMP6f response of a minimal size that must occur before propagation towards the soma begins. The data associated with varying stimulus parameters could potentially be useful here and reveal stimulus intensity/duration-dependent differences.

This is a very important point. It is difficult to pinpoint the beginning of the signal, which is why we rely on the average of responses. The additional analysis we provide based on temporal maps (new Figure 5) shows a very interesting result in that there is no significant difference between the spatial clustering of, or average distance between, activated domains in subthreshold and pre-soma suprathreshold activity. This result, along with the General Linear Model, suggests that there is not another subcellular potential spatial threshold, as the activity is the same. Instead, the main difference between activity in the domains that leads to soma activation or not is the overall percentage of domains active and not necessarily how that spatial activity is organized. We have also added this point in the Discussion section to highlight the importance of this result. P15, L3-8:

“Close examinations of the calcium surge uncovered distinct propagations whether before or after soma activation. Firstly, our analysis found that temporal clustering changed before and after calcium surge, with both being above subthreshold activity, and that this characteristic was absent when assessing spatial clustering. When comparing the percent area, spatial and temporal clustering of active domains using a GLM, we found that the percent area was the most significant parameter describing a threshold to soma activation.”

(2) Whether the propagation in the authors' experimental model is centripetal? This is implied throughout the manuscript but never shown. We think establishing whether (or not) the calcium dynamics are centripetal is important because it would clarify whether spatially adjacent domains within the "arbor" need to be sequentially active before reaching the threshold and then reaching the soma. More broadly, visualizing propagation will help to better visualize summation, which is presumably how the threshold is first reached (and overcome).

The alternative hypothesis of a general excitability threshold, as discussed above, would be challenged here and possibly rejected, thereby clarifying the nature of the Ca2+ process that needs to reach a threshold for further expansion to the soma and other parts of the astrocyte.

We agree that our view is centripetal when considering activity leading up to soma activation. Indeed, we have found arborization activity precedes soma activity (Figure 3), soma activity appears to rely on the percent area of domain activity (Figure 4), and pre-soma domain activity comes online earlier in domains distal from the soma (new Figure 5). However, whether this is intrinsic or due to the fact that synapses are more likely to occur in the periphery requires further studies. Our new results in the new Figure 5 demonstrating that subthreshold activity has a spatial organization that is not significantly different than pre-soma activity in suprathreshold cases argues in favor of a general excitability threshold hypothesis. However, we do not see these hypotheses as mutually exclusive. Excitingly, we have also found that following soma activation, calcium surge appears to follow a centrifugal propagation. We have since added the topic of a centripetal-centrifugal experimental model to the Discussion P15, L8-15:

“We then compared the delay of domain activation and its distance from the soma, and recreated previous results that suggest a centripetal model of astrocytic calcium responses from the distal arborizations to the soma (Fedotova et al., 2023; Rupprecht et al., 2024). Here, we went a step further and discovered that soma activation switches this directionality for astrocytic calcium surge to propagate outward in a centrifugal manner away from the soma. Taken together, these results demonstrate the integrative potential of astrocyte calcium responses and characterize further the astrocyte calcium surge to relay this other parts of the astrocyte.”

(3) In complement to the previous point: we understand that the spatial threshold does not per se have a location, but is there some spatial logic underlying the organization of active domains before the soma response occurs? One can easily imagine multiple scenarios of sparse heterogeneous GCaMP6f signal distributions that correspond to {greater than or equal to}22.6% of the arborization, but that would not be expected to trigger soma activation. For example, the diagram in Figure 4C showing the astrocyte response to 2Hz stim (which lacks a soma response) underscores this point. It looks like it has {greater than or equal to}22.6% activation that is sparsely localized throughout the arborization. If an alternative spatial distribution for this activity occurred, such that it localized primarily to a specific process within the arbor, would it be more likely to trigger a soma response?

This is an interesting point and our new spatiotemporal analysis found in the new figure (new Figure 5) aims to shed some light on this and is answered above. To our knowledge, there is no mechanism in astrocytes to impose directionality on calcium propagation, like rectifying voltage-gated sodium channels in neuronal voltage propagation. We found that the delay of domain activation compared to soma onset is significantly correlated to the distance from the soma (new Figure 5E). In addition, spatial clustering is not significantly different compared in pre-soma vs. non responders or post-soma. Together this suggests that centripetal propagation may be occurring throughout the entire cell and not in a local clustered way. Our findings also suggest that following soma activation astrocyte calcium surge follows a mostly centrifugal pattern (new Figure 5E).

(4) Does "pre-soma" activation predict the location and onset time of "post-soma" activation? For example, are arbor domains that were part of the "pre-soma" response the first to exhibit GCaMP6f signal in the "post-soma" response?

Please see above comments.

**Reviewer #2 (Public Review):**
Lines et al investigated the integration of calcium signals in astrocytes of the primary somatosensory cortex. Their goal was to better characterize the mechanisms that govern the spatial characteristics of calcium signals in astrocytes. In line with previous reports in the field, they found that most events originated and stayed localized within microdomains in distal astrocyte processes, occasionally coinciding with larger events in the soma, referred to as calcium surges. As a single astrocyte communicates with hundreds of thousands of synapses simultaneously, understanding the spatial integration of calcium signals in astrocytes and the mechanisms governing the latter is of tremendous importance to deepen our understanding of signal processing in the central nervous system. The authors thus aimed to unveil the properties governing the emergence of calcium surges. The main claim of this manuscript is that there would be a spatial threshold of ~23% of microdomain activation above which a calcium surge, i.e. a calcium signal that spreads to the soma, is observed. Although the study provides data that is highly valuable for the community, the conclusions of the current version of the manuscript seem a little too assertive and general compared with what can be deduced from the data and methods used.The major strength of this study is the experimental approach that allowed the authors to obtain numerous and informative calcium recordings in vivo in the somatosensory cortex in mice in response to sensory stimuli as well as in situ. Notably, they developed an interesting approach to modulating the number of active domains in peripheral astrocyte processes by varying the intensity of peripheral stimulation (its amplitude, frequency, or duration).

We thank the reviewer for their kind and thoughtful review of our study.

The major weakness of the manuscript is the method used to analyze and quantify calcium activity, which mostly relies on the analysis of averaged data and overlooks the variability of the signals measured. As a result, the main claims from the manuscript seem to be incompletely supported by the data. The choice of the use of a custom-made semi-automatic ROI-based calcium event detection algorithm rather than established state-of-the-art software, such as the event-based calcium event detection software AQuA (DOI: 10.1038/s41593-019-0492-2), is insufficiently discussed and may bias the analysis. Some references on this matter include: Semyanov et al, Nature Rev Neuro, 2020 (DOI: 10.1038/s41583-020-0361-8); Covelo et al 2022, J Mol Neurosci (DOI: 10.1007/s12031-022-02006-w) & Wang et al, 2019, Nat Neuroscience (DOI: 10.1038/s41593-019-0492-2). Moreover, the ROIs used to quantify calcium activity are based on structural imaging of astrocytes, which may not be functionally relevant.

Unfortunately, there is no general consensus for calcium analysis in the astrocyte or neuronal field, and many groups use custom made software made in lab or custom software such as GECIquant, STARDUST, AQuA or AQuA2. While AQuA is an event-based calcium event detection software, it may be that not including inactive domains that are SR101 positive could underestimate the spatial threshold for calcium surge. Our data is not based on the functional events but is based on calcium with structural constraints within a single astrocyte. This is crucial to properly determine the ratio of active vs inactive pixels within a single astrocyte.

For the reasons listed above, the manuscript would probably benefit from some rephrasing of the conclusions and a discussion highlighting the advantages and limitations of the methodological approach. The question investigated by this study is of great importance in the field of neuroscience as the mechanisms dictating the spatio-temporal properties of calcium signals in astrocytes are poorly characterized, yet are essential to understand their involvement in the modulation of signal integration within neural circuits.

We thank the reviewer for their suggestions to benefit the conclusions and discussion. We have now included a paragraph outlining the limitations of the study in the Discussion P15, L23-37:

“The investigation of the spatial threshold could be improved in the future in a number of ways. One being the use of state-of-the-art imaging in 3D(Bindocci et al., 2017). While the original publication using 3D imaging to study astrocyte physiology does not necessarily imply that there would be different calcium dynamics in one axis over another, the three-dimensional examination of the spatial threshold could refine the findings we present here. To better control the system, mice imaged here were under anesthesia, and this is a method that has been used to characterize many foundational physiological results in the field (Hubel and Wiesel, 1962; Mountcastle et al., 1957). However, assessing the spatial threshold in awake freely moving animals would be the next logical step. In this study, we chose to limit our examinations of calcium activity that was within the bounds determined by SR101 staining. Much work has shown that astrocyte territories are more akin to sponge-like morphology with small microdomains making up the end feet of their distal arborizations (Baldwin et al., 2024). Here, we took a conservative approach to not incorporate these fine morphological processes and only take SR101-postive pixels for analysis in order to reduce the possible error of including a neighboring astrocyte or extracellular space in our analyses. Much work can be done to extend these results.”

**Reviewer #3 (Public Review):**
Summary:The study aims to elucidate the spatial dynamics of subcellular astrocytic calcium signaling. Specifically, they elucidate how subdomain activity above a certain spatial threshold (~23% of domains being active) heralds a calcium surge that also affects the astrocytic soma. Moreover, they demonstrate that processes on average are included earlier than the soma and that IP3R2 is necessary for calcium surges to occur. Finally, they associate calcium surges with slow inward currents. Strengths:The study addresses an interesting topic that is only partially understood. The study uses multiple methods including in vivo two-photon microscopy, acute brain slices, electrophysiology, pharmacology, and knockout models. The conclusions are strengthened by the same findings in both in vivo anesthetized mice and in brain slices.

We thank the reviewer for the positive assessment of the study and his/her comments.

Weaknesses:The method that has been used to quantify astrocytic calcium signals only analyzes what seems to be a small proportion of the total astrocytic domain on the example micrographs, where a structure is visible in the SR101 channel (see for instance Reeves et al. J. Neurosci. 2011, demonstrating to what extent SR101 outlines an astrocyte). This would potentially heavily bias the results: from the example illustrations presented it is clear that the calcium increases in what is putatively the same astrocyte goes well beyond what is outlined with automatically placed small ROIs. The smallest astrocytic processes are an order of magnitude smaller than the resolution of optical imaging and would not be outlined by either SR101 or with the segmentation method judged by the ROIs presented in the figures. Completely ignoring these very large parts of the spatial domain of an astrocyte, in particular when making claims about a spatial threshold, seems inappropriate. Several recent methods published use pixel-by-pixel event-based approaches to define calcium signals. The data should have been analyzed using such a method within a complete astrocyte spatial domain in addition to the analyses presented. Also, the authors do not discuss how two-dimensional sampling of calcium signals from an astrocyte that has processes in three dimensions (see Bindocci et al, Science 2017) may affect the results: if subdomain activation is not homogeneously distributed in the three-dimensional space within the astrocyte territory, the assumptions and findings between a correlation between subdomain activation and somatic activation may be affected.

In order to reduce noise from individual pixels, we chose to segment astrocyte arborizations into domains of several pixels. As pointed out previously, including pixels outside of the SR101-positive territory runs the risk of including a pixel that may be from a neighboring cell or mostly comprised of extracellular space, and we chose the conservative approach to avoid this source of error. We agree that the results have limitations from being acquired in 2D instead of 3D, but it is likely to assume the 3D astrocyte is homogeneously distributed and that the 2D plane is representative of the whole astrocyte. Indeed, no dimensional effects were reported in Bindocci et al, Science 2017. We have included a paragraph in the discussion to address this limitation in our study on P15, L23-27:

“The investigation of the spatial threshold could be improved in the future in a number of ways. One being the use of state-of-the-art imaging in 3D(Bindocci et al., 2017). While the original publication using 3D imaging to study astrocyte physiology does not necessarily imply that there would be different calcium dynamics in one axis over another, the three-dimensional examination of the spatial threshold could refine the findings we present here.”

The experiments are performed either in anesthetized mice, or in slices. The study would have come across as much more solid and interesting if at least a small set of experiments were performed also in awake mice (for instance during spontaneous behavior), given the profound effect of anesthesia on astrocytic calcium signaling and the highly invasive nature of preparing acute brain slices. The authors mention the caveat of studying anesthetized mice but claim that the intracellular machinery should remain the same. This explanation appears a bit dismissive as the response of an astrocyte not only depends on the internal machinery of the astrocyte, but also on how the astrocyte is stimulated: for instance synaptic stimulation or sensory input likely would be dependent on brain state and concurrent neuromodulatory signaling which is absent in both experimental paradigms. The discussion would have been more balanced if these aspects were dealt with more thoroughly.

Yes, we agree that this is a limitation, and we acknowledge this is in the Discussion P15, L27-31:

“To better control the system, mice imaged here were under anesthesia, and this is a method that has been used to characterize many foundational physiological results in the field (Hubel and Wiesel, 1962; Mountcastle et al., 1957). However, assessing the spatial threshold in awake freely moving animals would be the next logical step.”

The study uses a heaviside step function to define a spatial 'threshold' for somata either being included or not in a calcium signal. However, Fig 4E and 5D showing how the method separates the signal provide little understanding for the reader. The most informative figure that could support the main finding of the study, namely a ~23% spatial threshold for astrocyte calcium surges reaching the soma, is Fig. 4G, showing the relationship between the percentage of arborizations active and the soma calcium signal. A similar plot should have been presented in Fig 5 as well. Looking at this distribution, though, it is not clear why ~23% would be a clear threshold to separate soma involvement, one can only speculate how the threshold for a soma event would influence this number. Even if the analyses in Fig. 4H and the fact that the same threshold appears in two experimental paradigms strengthen the case, the results would have been more convincing if several types of statistical modeling describing the continuous distribution of values presented in Fig. 4E (in addition to the heaviside step function) were presented.

We agree with the reviewer and have added to the paper a discussion for our justification on the use of the Heaviside step function, and have included this in the methods section. We chose the Heaviside step function to represent the on/off situation that we observed in the data that suggested a threshold in the biology. We agree with the reviewer that Fig. 4G is informative and demonstrates that under 23% most of the soma fluorescence values are clustered at baseline. We agree that a different statistical model describing the data would be more convincing and confirmed the spatial threshold with the use of a confidence interval in the text and supported the use of percent domains active for this threshold over other properties such as spatial or temporal clustering using a general linear model. P18-19, L34-2:

“Heaviside step function

The Heaviside step function below in equation 4 is used to mathematically model the transition from one state to the next and has been used in simple integrate and fire models (Bueno-Orovio et al., 2008; Gerstner, 2000).

H(a):={0,a<aτ1,a≥aτ

The Heaviside step function 𝐻(𝑎) is zero everywhere before the threshold area (𝑎) and one everywhere afterwards. From the data shown in Figure 4E where each point (𝑆(𝑎)) is an individual astrocyte response with its percent area (𝑎) domains active and if the soma was active or not denoted by a 1 or 0 respectively. To determine 𝑎 in our data we iteratively subtracted 𝐻(𝑎) from 𝑆(𝑎) for all possible values of 𝑎 to create an error term over 𝑎. The area of the minimum of that error term was denoted the threshold area.”

The description of methods should have been considerably more thorough throughout. For instance which temperature the acute slice experiments were performed at, and whether slices were prepared in ice-cold solution, are crucial to know as these parameters heavily influence both astrocyte morphology and signaling. Moreover, no monitoring of physiological parameters (oxygen level, CO2, arterial blood gas analyses, temperature etc) of the in vivo anesthetized mice is mentioned. These aspects are critical to control for when working with acute in vivo two-photon microscopy of mice; the physiological parameters rapidly decay within a few hours with anesthesia and following surgery.

We have increased the thoroughness of our methods section. Especially including that body temperature and respiration were indeed monitored throughout anesthesia.

**Recommendations for the authors:**

**Reviewer #1 (Recommendations For The Authors):**
(1) We think it would improve the paper if the authors provided a frame-by-frame example over (for example) 10-15 frames showing the spatiotemporal evolution of responses, where each frame represents 1s or 2s. This could be included with the temporal maps we proposed above.

We agree that this is a useful example and have included it in our new figure (new Figure 5, specifically see Figure 5A) that uses temporal maps to analyze the spatiotemporal properties of calcium dynamics (Figure 5B).

(2) Concerning the evidence in the present manuscript, we are not clear on what "populations" means. Can the authors clarify in methods? It is our understanding that 987 astrocytes from 30 populations from 3 mice were the source for the core data in the paper. What are the 30 populations, and how were the 987 astrocytes distributed across the populations? Are they roughly 10 FOVs per mouse? If so, please clarify roughly how far apart FOVs from the same mouse were, and how much delay between stim protocol application there was when a FOV was changed to a new FOV. Also, if for example, the 10th FOV from mouse 1 "saw" 9 rounds of stimulation before recording the response to the 10th stim round. To this point, was there any indication of response differences in populations that were recorded earlier vs later in the experimental sequence for each mouse?

Descriptions of data will be included with the uploaded datasets following acceptance.

(3) The description of the results on page 6 is a bit confusing for us. In lines 1-4, are the authors saying that 57.7% of astrocytes in a FOV exhibited responses within their soma and arborization, while 15.1% had responses only in arborization? If so, this is not clear to us from Figure 2C, where we count ~25 astrocytes in the FOV, maybe 8 or 9 astrocytes with activity in the arborization + soma (after stimulation), and 8 or 9 astrocytes with responses only in arborization. Is there something we do not understand, or is the second panel simply not representative of the group data?

Figure 2D is representative of the group data and does indeed show 57.7% of the population responds within the soma and arborization, and a 15.1% of astrocytes with responses in only their arborizations. It is unable to observe in this image whether arborizations are active or just increases in one or a few domains, as may not be enough activity to be detected when sampling over the entire arborization.

(4) In the second part of page 6 - when the authors apply linear regression - are they saying that there is a linear relationship between the amount (area) of activity measured in the arborization versus the soma, where populations of astrocytes with 50% activation of the arborization also tend to have 50% activation in their somas? If so, then this is not apparent by the map provided in Figure 2C, where it looks like soma activation (within the subpopulation) is 100% irrespective of the apparent activity in the arborization. This needs to be clarified. If not, and what they mean is that the probability of finding an active soma is related to the amount of activation within the arborization, this needs to be stated more clearly.

When testing the linear relationship between somas active vs arborizations active, we find a significant linear correlation (p < 0.001, R2 = 0.90).

(5) In the experiments where stimulation duration, frequency, and intensity were varied to determine the percentage of domains that were on, it would be helpful to better understand the protocol in terms of sequence. In the methods it seems that hindpaw stimulation intensity was first pseudo-randomly varied at 2Hz for 10s, followed by pseudorandomly varied stimulation frequency and then pseudo-randomly varied duration - both at 2mA for 10s. Is this correct?

We have since updated the methods section to better describe the experimental protocol.

(6) In Figure 3E the alignment of the "arbor" to the somatic response is a bit misleading. The signals being averaged for the "arbor" are composed of temporally heterogeneous sources (from distal and proximal domains) and when averaged will produce an artificially slow rise time. In contrast, the averaged somatic signals are composed of much more homogenous sources (arising from a more singular event) and therefore have a sharp rise time. It would make more sense to align their kinetics relative to the stimulus onset. It would also make more sense to compare the somatic response of astrocytes to the "arbor" of astrocytes which respond rapidly vs slowly to the foot-shock.

Aligning the responses to the stimulus onset would exacerbate the artificially slow rise time for the soma and arborization as not all cells come online at the same time from stimulus onset.

**Reviewer #2 (Recommendations For The Authors):**
Data availabilityIt seems that the data is not shared on a public repository, while it appears to be necessary according to eLife's general principles (see https://elife-rp.msubmit.net/html/eliferp_author_instructions.html#dataavailability).

We will upload raw data to a repository upon acceptance of the manuscript.

Data analysis- Why did the authors choose the heaviside step function to characterize conditions for somatic event initiation? It seems that this approach is averaging very heterogeneous data (some cells do not display somatic events even with ~50% domains active while some display somatic events with < 5 it seems).

Please see discussion to variability in the responses to the public reviews. We have since included more discussion on the use of the Heaviside step function in the Methods section.

- Averaging of the data. It seems that the approach chosen to quantify calcium activity overlooks the variability of the signals measured ("Astrocyte calcium quantifications were averaged over all astrocytes of a single video and these values were used in statistical testing.", l.22-23, page 15). What is the variability of the measured features between different astrocytes? Between different animals? To what extent does this averaging strategy overlook the variability of the signals/how much information do we expect to lose? The manuscript would probably benefit from a more advanced statistical approach to analyze the data.Is it possible to extract information from the data that would indicate mechanisms allowing somatic activity when the percentage of domain activation was lower than the threshold? How about the opposite (i.e when no global event was triggered even when the percentage of domain activation was high)?

We are indeed combining the responses from many different diverse astrocyte responses, and we see this as a strength of the paper. Variation is a hallmark of biology, and we have added this to the discussion. In the rare cases where astrocyte somas do not come online when the percent of arborizations is over threshold, or the opposite when somas activate with little domain activation, we would say this is most likely due to imaging 2D instead of the entire 3D cell. We have also added this into our discussion.

- Here are a few suggestions for additional analysis that might be of interest to the community:- Measuring calcium activity in domains depending on their distance from the soma. This would allow us to better understand the spatial integration of the signals and notably answer the following question: Does the emergence of somatic events depend on the spatial distribution of active domains? (and does a smaller domain-soma distance facilitate the emergence of a calcium surge with a lower percentage of active domains?) These measurements could be visualized with plots of xy position of the domains (domain-soma distance) = f(time) with a colormap reflecting dF/F0, for example, at different times pre- and post-somatic events. Instead of DF/F0, these plots could also display the correlation between domain activities.

We have performed this analysis, and it is now in the new figure (new Figure 5).

- Adding temporality to the data analysis. It seems that calcium activity is "concatenated" during the whole duration prior to the somatic event (pre-soma) and after (post-soma). However, it is unclear how long the domains remained active and how many domains were still active at the onset of the somatic event. Adding a finer temporal analysis might help answer questions such as the potential need for some degree of synchronization of domain activity to trigger calcium surges.It could notably be interesting to measure the level of synchrony of events as a function of their distance from the soma and to analyze how it correlates with the properties of the somatic event.

We have now included temporal analysis of astrocyte calcium surge in our new figure (new Figure 5). While we did see examples of spatially clustered domain activation in our data, those examples usually included other non-clustered domain activities and when including all of the active domains within an astrocytes arborization, we found no difference between the distance between activated domains before and after soma activation, even when comparing to subthreshold domain activity.

Experiments- Would it be possible to apply different levels of stimulation to a given cell in order to discriminate whether the "no-soma" cells can display somatic events when neuronal activity is enhanced?

Increased sensory stimulation does increase soma activity (Please see Lines et al., Nature Communications, 2020). An example of increased stimulation leading to somatic activation where it was not present in lower stimuli can be seen in Figure 4A-C.

- Why choose a stimulation of 2 mA, 2 Hz for 20 sec in the experiments on IP3R2-/- mice?Has the same set of various stimulation protocols featured in Figure 4 been applied to IP3R2-/- mice? If so, were more domains activated as stimulation intensity (amplitude; duration, or frequency) increased? Could it trigger somatic events? This information seems necessary to be able to assert that calcium surges rely on the IP3R2 pathway.

These experiments were not performed.

- Adding intermediary values of ATP pulse duration to Figure 6 (e.g. 50 ms and 75 ms) might strengthen the claim that the linear increase of SIC frequency with ATP application duration is only observed above the ~23% threshold.

Agreed, however these experiments were not performed.

Minor corrections to the text and figures.MethodsThe reader might benefit from a little more detail regarding the analysis of calcium signals. Notably, what was the duration of the calcium recordings? Was it constant across the different conditions tested in the study? Was it different in slice experiments versus in vivo experiments? What were the durations of the pre- and post- soma recordings and their variability? Was the calcium activity normalized for each astrocyte or animal? If not, why not consider normalizing the post-stimulation activity with pre-stimulation baseline activity?Similarly, some information on the stimulation protocol seems to be lacking: what was the frequency and intensity of the stimulus in the experiments where stimulus duration varied? Concurrently, what were the duration and intensity when frequency varied? What were the duration and frequency when the intensity varied?It might be beneficial to add further information on the algorithm of the Calsee software. What is it performing? How was it tested? Why is it referred to as "semi"-automatic, i.e. what might the user be needing to do manually? The segmentation seems to be omitting some branches connecting distal ROIs to the soma (see e.g. Fig S1.E). How would this influence the analysis and results?Results- Some assessments in the manuscript seem a bit too assertive/general compared to what can be deduced from the evidence presented in the figures. It could be beneficial to the reader to rephrase the latter. Some examples are listed below:- "These results indicate that astrocyte responses occurred initially in the arborizations, which is consistent with the idea that synapses are likely to be accessed at the astrocyte arborization ", l.11-12 page 7. The fact that the time to peak is lower in the arborization does not necessarily mean that signals initiate there. It could be because the kinetics/pathways in those compartments are different or there could be a dilution effect in the soma. Indeed, an influx of the same amount of calcium ions in the soma vs in a small domain will not correspond to the same DF/F0 in those compartments and might thus remain undetected in the soma.- "Using transgenic IP3R2-/- mice, we found that the activation of type-2 IP3 receptors is necessary for the generation of astrocyte calcium surge" (page 4, line 1-2), "present data further demonstrate that IP3R2 are necessary for the propagation of astrocyte calcium surge." (l. 18-19 page 13) -> As discussed above, the evidence does not seem to be strong enough to assert that IP3R2 is necessary to trigger somatic events. The results indicate that the IP3R2 pathway seems to facilitate the emergence of somatic events. As astrocytes differ strongly in terms of morphology and expression profiles depending on physiological conditions, the conclusions of this study might only apply to the specific experimental conditions used: region studied, age of the animal, type of sensory stimuli performed, and so on.- "These results indicate that spatial threshold of the astrocyte calcium surge has a functional impact on gliotransmission, which have important consequences on the spatial extension of the astrocyte-neuron communication and synaptic regulation", l.41-48 page 11. Figure 6 seems to indicate a correlation between the proportion of astrocyte domains activated and the frequency of SICs. The data seems insufficient to conclude that there is a causal relationship between calcium surge in the astrocyte and gliotransmission or SIC frequency.-" These results indicate that, on average, subcellular calcium events located in astrocyte arborizations are related to soma activation.", page 6 l 15-16. It may be more informative to specify the correlation measured: i.e the larger the arborization activity, the larger the percentage of active somas.FiguresFigure 2: Adding more details in the figure legend explaining how the different parameters are calculated might be useful to the reader. Notably, what does soma active (%) refer to?Figure 3: Could it be possible to add individual traces of calcium activity in the soma and arborization of individual cells to provide a glimpse of the variability of the signals measured?Fig4. B-C: Could it be possible to add in the legend information on the timeline between stimulation and calcium signal recording? (and the duration of the latter).Fig4 D-E: Why is the maximum number of active domains in panel D ~50-60% but goes up to ~100% in panel E? Could it be that plotting SEM rather than STD might misrepresent the variability in the percentage of active domains for each stimulus property?Fig4F: It seems that the threshold changes with the frequency of the stimulus: e.g. at 10 Hz, the threshold seems larger than 22.6%. What would that mean?Fig4G: - Why do some data points display a soma amplitude < 0 DF/F0 ?- Why choose a sigmoid fit? What are the statistics associated to the fit? Is it in accordance with the threshold of 23%? Would a linear fit provide a good fit?Fig5F: - It seems that a few IP3R2-/- astrocytes displayed somatic events? If so, it might be interesting to mention this in the discussion section and to speculate on why that might be. - It seems that panel 5F displays the average percentage of somas that got activated rather than the probability of somatic events.- Is it possible that the effect seen in domains vs arborization is due to statistical effects (as n=2450 vs 112)?Fig S1: Panel D legend: double labeling of the radius used for each plot might be useful, notably for colorblind readers as the colors might be hard to see.Discussion- The discussion section might benefit from a discussion on the similitude between the data presented here and previous reports that reported similar results, i.e that most calcium signals in astrocytes were located in the distal processes, forming microdomains that rarely propagated to the soma. These include Bindocci et al 2017 Science (DOI:10.1126/science.aai8185) and Georgiou et al, Science Advances, 2022 (DOI: 10.1126/sciadv.abe5371).

Thank you for the suggestions. We have now changed portions of the Methods, Results and Discussion sections.

**Reviewer #3 (Recommendations For The Authors):**
The text could potentially be improved somewhat.

Thank you.